# Scalable Multilingual Multimodal Machine Translation with Speech-Text Fusion

**Yexing Du**[1,2], **Youcheng Pan**[2*], **Zekun Wang**[1], **Zheng Chu**[1], **Yichong Huang**[1]
**Kaiyuan Liu**[1,2], **Bo Yang**[2], **Yang Xiang**[2*], **Ming Liu**[1,2*], **Bing Qin**[1,2]

[1]Harbin Institute of Technology    [2]Pengcheng Laboratory

## Abstract

Multimodal Large Language Models (MLLMs) have achieved notable success in enhancing translation performance by integrating multimodal information. However, existing research primarily focuses on image-guided methods, whose applicability is constrained by the scarcity of multilingual image-text pairs. The speech modality overcomes this limitation due to its natural alignment with text and the abundance of existing speech datasets, which enable scalable language coverage. In this paper, we propose a **Speech-guided Machine Translation (SMT)** framework that integrates speech and text as fused inputs into an MLLM to improve translation quality. To mitigate reliance on low-resource data, we introduce a **Self-Evolution Mechanism**. The core components of this framework include a text-to-speech model, responsible for generating synthetic speech, and an MLLM capable of classifying synthetic speech samples and iteratively optimizing itself using positive samples. Experimental results demonstrate that our framework surpasses all existing methods on the Multi30K multimodal machine translation benchmark, achieving new state-of-the-art results. Furthermore, on general machine translation datasets, particularly the FLORES-200, it achieves average state-of-the-art performance in 108 translation directions. Ablation studies on CoVoST-2 confirms that differences between synthetic and authentic speech have negligible impact on translation quality. The code and models are released at https://github.com/yxduir/LLM-SRT.

## 1 Introduction

Multimodal Machine Translation (MMT) leverages complementary information from multiple modalities, such as images, to enhance machine translation (MT) quality. These modalities provide supplementary contextual information for source texts, thereby mitigating ambiguities caused by polysemy or omissions (Shen et al., 2024).

Traditionally, image-based MMT models (Cheng et al., 2024) process image-text pairs to generate translations, leveraging visual context for semantic disambiguation. However, these models require an associated image for each input text, which limits their applicability. Recent image-free approaches (Guo et al., 2023) have employed diffusion models (Rombach et al., 2022) to generate synthetic images to enhance translation. While these studies address the issue of image dependency, those methods still face two limitations: (1) **Generalizability**: While MMT models perform well on ambiguous datasets (Elliott et al., 2016), they struggle to generalize to general translation datasets and even introduce noise in some scenarios (see Figure 1). (2) **Multilinguality**: Existing image MMT datasets (Guo et al., 2022) support only a few languages, with limited of languages coverage (see Table 1). Advances in diffusion Text-to-Speech (TTS) models (Du et al., 2024) have achieved high-quality, zero-shot multilingual speech synthesis. This raises a question: **Can we leverage speech modalities to enhance translation quality**?

Recent studies have revealed that, alongside lexical information, speech signals also convey prosodic cues, which offer valuable supplementary information (Chi et al., 2025). Inspired by fusion of text and prosody features, we propose the framework of Speech-guided Machine Translation (SMT),

---

*Corresponding Authors: {panych,xiangy}@pcl.ac.cn,mliu@ir.hit.edu.cn

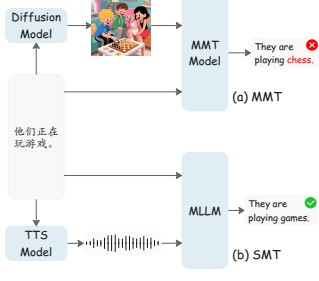

| Image Dataset | Language | Speech Dataset | Language |
|---|---|---|---|
| IAPR TC-12 (Grubinger, 2006) | eng, deu | MuST-C (Di Gangi et al., 2019) | eng→15 |
| Multi30K (Elliott et al., 2016) | eng, deu, deu, fra | CoVoST-2 (Wang et al., 2020) | 21→eng eng→15 |
| MLT (Lala & Specia, 2018) | eng, deu | Europarl-ST (Iranzo-Sánchez et al., 2020) | 9↔9 |
| MultiSense (Gella et al., 2019) | eng, deu, spa | Fleurs (Conneau et al., 2022) | 102↔102 |
| AmbigCaps (Li et al., 2021) | eng, tur | Granary (Koluguri et al., 2025) | 25→eng |
| Fashion-MMT (Song et al., 2021) | eng, cmn | CCFQA (Du et al., 2025a) | 8↔8 |
| EMMT (Zhu et al., 2023) | eng, cmn | BhasaAnuvaad (Sankar et al., 2025) | eng↔14 |
| TIT Dataset (Ma et al., 2022) | eng, deu, cmn | | |
| BLATID (Chen et al., 2023) | eng, cmn | | |
| OCRMT30K (Lan et al., 2023) | eng, cmn | | |
| MSCTD (Liang et al., 2022) | eng, deu, cmn | | |
| BIG-C (Sikasote et al., 2023) | eng, ben | | |
| HaVQA (Parida et al., 2023) | eng, hau | | |
| M³ (Guo et al., 2022) | eng→6 | | |

Figure 1: Image-Guided vs. Speech-Guided Machine Translation.

Table 1: Dataset Statistics. For the languages supported by the image datasets, please refer to Table 7. Our MLLM supports 28 languages, as shown in Table 8.

which maps speech-text fusion inputs {*speech*, *text*} to {*translation*} outputs. Specifically, our SMT framework integrates a TTS model with an MLLM through a self-evolution mechanism (Tao et al., 2024) that leverages synthetic speech to enhance translation performance.

The framework consists of two core components: (1) **MLLM Pre-training**: We employ a multi-stage curriculum learning strategy with progressively complex objectives, beginning with speech recognition (ASR) for speech-text mapping, then speech-to-text translation (S2TT) for cross-lingual and cross-modality bridging, and culminating in SMT training for joint speech-text processing. (2) **Self-Evolution Mechanism**: This component synthesizes training data via the TTS model, where the MLLM classifies speech samples based on translation scores. The MLLM undergoes continuous training using positive samples, while translation performance metrics serve as evolution objectives, enabling continuous framework improvement through iterative refinement cycles.

The experimental results demonstrate that our framework achieves new state-of-the-art (SOTA) results on the Multi30K benchmark (Elliott et al., 2016), surpassing all existing MMT approaches. Our framework further achieves SOTA average machine translation (MT) performance across 108 languages directions on the FLORES-200 benchmark (Team et al., 2022), outperforming much larger language models. Ablation studies on the CoVoST-2 dataset (Wang et al., 2020) also reveal that the discrepancy between synthetic and authentic speech has a negligible effect on translation performance. In summary, our key contributions are as follows:

- We propose a novel speech-guided machine translation framework, which consists of a TTS model and an MLLM. Our framework leverages prosodic cues in speech to enhance translation performance and supports 28 languages.
- We propose a self-evolution framework that autonomously generates training data for iterative self-enhancement. The framework employs continual training for the MLLM, utilizing synthetic data to improve the model's low-resource translation quality.
- Our framework achieves state-of-the-art results on MMT and MT tasks across multiple benchmarks (Multi30K, FLORES-200). Ablation studies on the CoVoST-2 benchmark show that the difference between authentic and synthetic speech has a negligible impact on translation performance.

## 2 METHODOLOGY

### 2.1 MODALITY-AGNOSTIC HYPOTHESIS

**Assumption 1.** Any auxiliary modality can enhance machine translation performance when:

- The modality provides semantically relevant information to the source text.
- The modality representation can be aligned and jointly optimized with textual features in a shared latent space, given sufficient training data to learn discriminative embeddings.

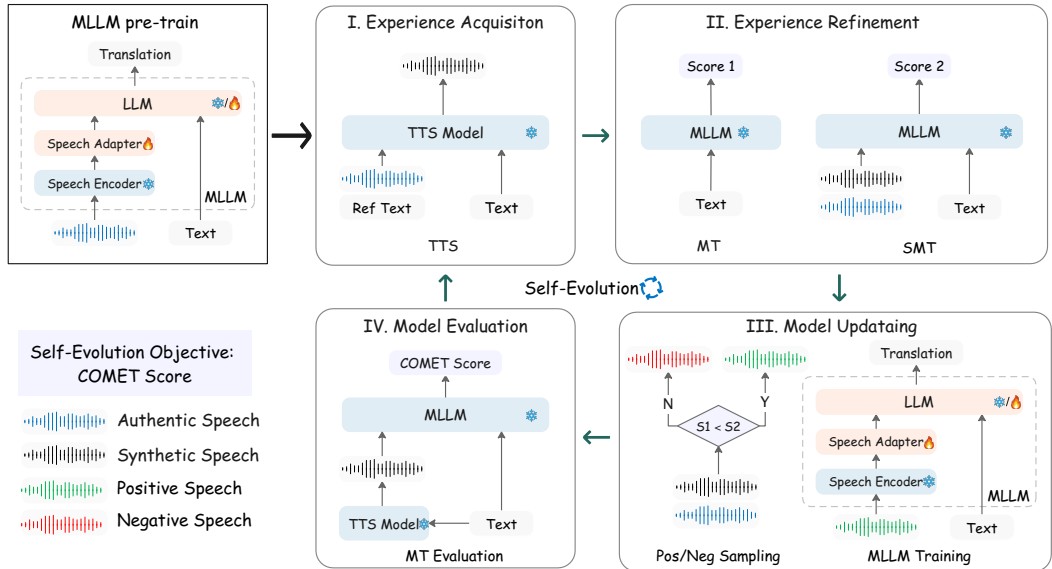

Figure 2: Overview of Our SMT Framework. The proposed system architecture comprises two core components: (1) MLLM pretraining and (2) Self-Evolution. This framework takes text input, synthesizes speech of the text via a TTS model, and leverages the MLLM to process both text and speech features for higher-quality translation output. Self-evolution mechanism can autonomously generate training data to iteratively optimize the framework.

## 2.2 OVERALL DESIGN

Figure 2 illustrates the SMT framework, comprising an MLLM and a TTS model. The processing pipeline operates as follows: First, the system accepts textual input and synthesizes speech via the TTS model. Then, the MLLM processes both the text and synthetic speech to generate translations. The following subsections detail two key components: MLLM pretraining (Section 2.3) and self-evolution mechanism (Section 2.4).

## 2.3 MLLM PRE-TRAINING

The MLLM is built upon a large language model (LLM) (Cui et al., 2025), adopts Whisper's encoder (Radford et al., 2023) as the speech encoder, followed by a Q-Former (Li et al., 2023a) and MLP layer for speech adapter. We design a three-stage training pipeline and perform instruction tuning. The sequential fine-tuning stages comprise: (1) automatic speech recognition, (2) speech-to-text translation, and (3) speech-guided machine translation.

**ASR.** The MLLM learns speech-text alignment through ASR pre-training while keeping only the speech adapter trainable.

**S2TT.** Given speech input and instructions, the MLLM simultaneously generates transcriptions and translations.

**SMT.** The MLLM processes joint speech-text inputs to generate translation outputs by leveraging complementary multimodal information.

| Modules | Param | Stage | Details |
|---|---|---|---|
| Speech Encoder | ~635M | - | Whisper's encoder |
| Speech Adapter | ~80.5M | All | Q-Former and MLP |
| LLM | ~9.2B | - | GemmaX2-28-9B |
| LLM adapter | ~8.9M | III | LoRA (r=16, alpha=32) |
| Total | ~10B | | |

Table 2: MLLM Pre-training. The blue color indicates the number of trainable parameters.

## 2.4 Self-Evolution Mechanism

Self-evolution mechanism allows models to autonomously learn through four phases: experience acquisition, experience refinement, updating, and evaluation. Our SMT framework is based on (1) MLLM, (2) TTS model, and (3) a S2TT dataset with authentic speech, text, and translation.

### 2.4.1 Stage I: Experience Acquisition

The purpose of this stage is to generate synthetic speech. During this stage, the prompt text and the predicted speech duration are strictly aligned with authentic speech and text pairs.

**TTS Inference.** We employ a TTS model to synthesize speech signals from the text in the S2TT dataset. Given a reference text, the TTS model generates a new speech utterance while cloning a randomly selected voice from the same dataset. This process ensures a diverse set of synthetic speech data with varied prosody, which is crucial for our framework's training.

### 2.4.2 Stage II: Experience Refinement

This stage implements a quality-aware labeling strategy for speech samples. We find that not all speech is beneficial for translation, so we need to classify the samples. This process is achieved by comparing the scores of MT and SMT.

**MT and SMT Inference.** The MLLM operates in two distinct modes. In MT mode, the model processes textual inputs $t_{text}$ to generate translations $t_{trans}$, producing score $S_1$. In SMT mode, the model accepts either authentic speech $s_{ref}$ or synthetic speech $s_{gen}$ paired with its corresponding text input to generate translations, producing score $S_2$.

### 2.4.3 Stage III: Model Updating

This stage is dedicated to optimizing the MLLM by leveraging the synthetic data generated in the previous stage. The primary goal is to enhance the MLLM's ability to effectively utilize prosodic cues from speech input for improved translation quality.

**Positive/Negative Sampling.** We first perform a comparative analysis to categorize each synthesized speech-text pair into either a positive ($s_{pos}$) or a negative ($s_{neg}$) sample. Let $S_1$ be the translation quality score with text input only, and $S_2$ be the score when the MLLM receives both text and speech input.

A sample is categorized as a **positive sample** ($s_{pos}$) if the additional speech input improves translation performance ($S_2 > S_1$). Conversely, a sample is labeled as a negative sample ($s_{neg}$) if the speech input provides no benefit ($S_2 \leq S_1$). The scores are computed as:

$$\begin{cases} S_1 = \text{COMET}\Big(\text{MLLM}\big(t_{text}\big), t_{trans}\Big) \\ S_2 = \text{COMET}\Big(\text{MLLM}\big(s_{ref} \text{ or } s_{gen}, t_{text}\big), t_{trans}\Big) \end{cases} \quad (1)$$

**MLLM Continuous Training.** The MLLM is then continually fine-tuned using only the identified positive samples ($s_{pos}$). This targeted training strategy guides the model to prioritize and learn from the most beneficial speech-text interactions, thereby enhancing its ability to leverage prosody for superior translation performance.

### 2.4.4 Stage IV: Model Evaluation

In this final stage, we evaluate the framework's translation performance to determine whether to continue the self-evolution loop. We synthesize speech for the evaluation text using a fixed reference voice and measure the SMT framework's performance with the COMET score. This process iterates until the COMET score on the evaluation set converges and no longer shows significant improvement.

## 3 EXPERIMENTS

### 3.1 DATASETS

We conduct comprehensive evaluations on several benchmarks. For multimodal machine translation, we use Multi30K[1] (Elliott et al., 2016). For machine translation, we use FLORES-200[2] (Team et al., 2022) and WMT24++[3] (Deutsch et al., 2025). Additionally, we perform ablation studies on the CoVoST-2[4] dataset (Wang et al., 2020). Detailed information for datasets is provided in Table 10.

### 3.2 EXPERIMENT SETUP

**Model Architecture.** Our MLLM consists of a frozen speech encoder, specifically the encoder from Whisper-large-v3 (Radford et al., 2023), and a trainable adapter layer. This adapter comprises a Q-Former (Li et al., 2023b) and a multilayer perceptron (MLP). The LLM backbone is GemmaX2-28-9B (Cui et al., 2025). Following the configuration in (Yu et al., 2024), our Q-Former uses 80 queries, each with a dimension of 768. The datasets used for MLLM training are detailed in Table 9. For the TTS model, we adopt the CosyVoice2 (Du et al., 2024) model.

**Training Details.** Experiments are conducted on four A100 GPUs (80GB). Following the experimental setup (Ma et al., 2026), we used the AdamW optimizer (Loshchilov, 2017) with a peak learning rate of $1 \times 10^{-4}$. The learning rate was linearly warmed up over 1K steps and then linearly decayed for the remainder of the training. The models can be trained in under a week.

**Evaluation Metrics.** For evaluation, we employ BLEU[5] (Post, 2018), spBLEU (Team et al., 2022), and COMET[6] (Rei et al., 2020). We compute spBLEU using the tokenizer "flores200". For a fair comparison, our LLM inference uses vLLM (Kwon et al., 2023), with all beam search settings and temperature uniformly set to 1 and 0, respectively.

### 3.3 COMPARING MODELS

**MT Models.** We evaluate the translation performance of four models: Deepseek-V3.1 API (Guo et al., 2025), Gemma3-27B-it (Team et al., 2025), Qwen3-Next-80B-A3B-Instruct (Team, 2024), and NLLB-54B (Team et al., 2022).

**MMT Models.** We compare our framework against two categories of existing multimodal machine translation models. We compare against four traditional MMT models that use text and authentic image: Soul-Mix (Cheng et al., 2024), RG-MMT-EDC (Tayir & Li, 2024), WRA-guided (Zhao et al., 2022), and ConsQA-MMT (Gao et al., 2025b). Additionally, we compare against four image-free MMT models that rely on text and synthetic image: VALHALLA (Li et al., 2022), Bridge (Guo et al., 2023), DreamLLM (Dong et al., 2024), and IMAGE (Chen et al., 2024a).

### 3.4 OVERALL RESULTS

Our comprehensive experiments demonstrate the significant effectiveness of our proposed speech-guided machine translation approach. Our framework achieves new state-of-the-art results on the Multi30K benchmark, surpassing traditional text-only and image-based MMT models. SMT-9B also consistently outperforms much larger text-only language models. Furthermore, our framework shows strong generalization, achieving state-of-the-art results in 108 translation directions on the FLORES-200 benchmark. Finally, ablation studies confirm that the performance difference between authentic and synthetic speech is negligible.

---

[1] https://github.com/multi30k/dataset
[2] https://github.com/facebookresearch/flores
[3] https://huggingface.co/datasets/google/wmt24pp
[4] https://github.com/facebookresearch/covost
[5] https://github.com/mjpost/sacrebleu
[6] https://huggingface.co/Unbabel/wmt22-comet-da

| Models | eng → deu | | | eng → fra | | | eng → ces | |
|---|---|---|---|---|---|---|---|---|
| | Test2016 | Test2017 | MSCOCO | Test2016 | Test2017 | MSCOCO | Test2016 | Test2018 |
| **Models based on Text** | | | | | | | | |
| DeepSeek-V3.1 (Guo et al., 2025) | 44.2 / 87.3 | 41.1 / 86.8 | 36.4 / 83.2 | 55.3 / 88.2 | 54.0 / 87.7 | 53.5 / 85.8 | 37.9 / 90.7 | 35.9 / 89.7 |
| Gemma3-27B-it (Team et al., 2025) | 43.7 / 87.1 | 40.3 / 86.3 | 36.1 / 83.2 | 55.4 / 87.9 | 54.3 / 87.9 | 49.6 / 85.0 | 36.4 / 89.9 | 35.9 / 89.1 |
| NLLB-moe-54B (Team et al., 2022) | 41.4 / 86.2 | 39.7 / 85.8 | 34.7 / 82.1 | 55.1 / 87.4 | 54.8 / 87.7 | 53.3 / 85.3 | 35.7 / 88.9 | 35.8 / 88.3 |
| Qwen3-Next-80B-A3B (Team, 2025) | 41.6 / 86.3 | 37.6 / 85.9 | 31.9 / 82.5 | 53.2 / 87.8 | 51.9 / 87.6 | 50.4 / 85.1 | 29.2 / 87.2 | 27.9 / 85.9 |
| **Models based on Text & Authentic Image** | | | | | | | | |
| WRA-guided † (Zhao et al., 2022) | 39.3 / —— | 32.3 / —— | 28.5 / —— | 61.8 / —— | 54.1 / —— | 43.4 / —— | —— | —— |
| RG-MMT-EDC † (Tayir et al., 2024) | 42.0 / —— | 33.4 / —— | 30.0 / —— | 62.9 / —— | 55.8 / —— | 45.1 / —— | —— | —— |
| Soul-Mix † (Cheng et al., 2024) | 44.2 / —— | 37.1 / —— | 34.2 / —— | 64.7 / —— | 57.4 / —— | 49.2 / —— | 36.5 / —— | 32.8 / —— |
| ConsQA-MMT † (Gao et al., 2025a) | 44.2 / —— | 37.6 / —— | 34.3 / —— | 64.8 / —— | 58.3 / —— | 48.5 / —— | 34.7 / —— | 30.3 / —— |
| **Models based on Text & Synthetic Image** | | | | | | | | |
| VALHALLA † (Li et al., 2022) | 42.7 / —— | 35.1 / —— | 30.7 / —— | 63.1 / —— | 56.0 / —— | 46.5 / —— | —— | —— |
| Bridge † (Guo et al., 2023) | 42.5 / —— | 36.0 / —— | 32.0 / —— | 63.7 / —— | 56.2 / —— | 46.3 / —— | 35.2 / —— | 31.2 / —— |
| DreamLLM † (Dong et al., 2024) | 27.2 / 74.8 | 19.5 / 73.5 | 19.3 / 69.4 | 36.9 / 81.1 | 34.7 / 80.6 | 36.6 / 79.2 | —— | —— |
| IMAGE † (Chen et al., 2025) | 45.3 / 83.1 | 38.6 / 81.9 | 37.5 / 78.8 | 67.5 / 88.3 | 61.5 / 86.6 | 49.3 / 82.5 | —— | —— |
| **Models based on Text & Synthetic Speech** | | | | | | | | |
| Baseline (Text only) | 42.9 / 87.0 | 38.8 / 86.4 | 34.3 / 82.7 | 52.4 / 87.7 | 52.0 / 87.9 | 52.6 / 86.1 | 34.1 / 89.9 | 34.8 / 89.0 |
| Baseline + Lora (Text only) | 44.0 / 87.0 | 39.4 / 86.4 | 35.3 / 83.0 | 55.5 / 88.1 | 54.0 / 88.2 | 53.4 / 85.9 | 37.2 / 90.0 | 35.7 / 89.1 |
| SMT-9B | 47.0 / 87.8 | 41.8 / 87.3 | 38.5 / 84.0 | 67.0 / 90.0 | 62.1 / 89.6 | 55.3 / 86.7 | 41.4 / 90.8 | 39.9 / 89.8 |

Underlined denotes previous state-of-the-art models, while highlighted surpasses the previous models.

Table 3: Translation Performance on Multi30K (BLEU / COMET) MMT Benchmark. The average character length of the input English text is **59.3**. † indicates that the scores were directly cited from other research papers.

### 3.4.1 MAIN RESULTS FOR MULTIMODAL MACHINE TRANSLATION

**Comprehensive Performance Improvement from Speech-Text Fusion Input.** Table 3 showcases the remarkable performance of our SMT-9B model, which expertly integrates both synthetic speech and text inputs. The results clearly demonstrate a substantial performance gain across all evaluated test sets. Specifically, for the eng→deu task, our model attains impressive BLEU scores of 47.0, 41.8, and 40.3 on the Test2016, Test2017, and MSCOCO datasets, respectively. Similarly, for the eng→fra task, it achieves high BLEU scores of 67.0, 62.1, and 55.3. These scores consistently and significantly outperform all text-only baselines. The clear advantage our approach holds provides compelling evidence that synthetic speech, as an auxiliary modality, can furnish crucial prosodic and contextual information that is not available in text alone, thereby effectively enhancing machine translation performance.

**Competitive Advantage of Synthetic Speech in Multimodal Translation.** The table clearly demonstrates the significant performance advantage of our proposed method, which leverages synthetic speech, over existing multimodal machine translation models that primarily rely on visual inputs. Our SMT-9B model establishes a new benchmark by achieving a state-of-the-art average BLEU score of 52.0. This score not only surpasses the performance of all previous methods but does so by a substantial margin, regardless of whether those models used authentic or synthetic images. For a direct comparison, our model outperforms the best-performing image-based model by an impressive 2.1 points (which only achieved an average BLEU of 49.9). This result suggests that the speech modality is a rich and unique source of contextual information that is both distinct from and complementary to the visual modality.

**Comparative Analysis with Large-Scale Language Models.** Although not shown in the table, our SMT-9B model, despite having a parameter count that is only 1/67th of the DeepSeek-V3-671B model, achieves superior translation performance. This result highlights the significant potential of multimodal learning: even a smaller model can achieve or surpass the performance of a much larger text-only model by effectively leveraging cross-modal information. This demonstrates that modality fusion can compensate for a lack of scale, offering a viable path for developing high-performance translation systems in resource-constrained environments.

| Models | FLORES-200 | | | | WMT24++ | |
|---|---|---|---|---|---|---|
| | eng → 27 | jpn → 27 | kor → 27 | cmn → 27 | eng → 22 | eng → 22 (<200) |
| **Models based on Text** | | | | | | |
| DeepSeek-V3.1 (Guo et al., 2025) | 39.3 / 88.9 | 26.1 / 85.7 | 27.7 / 85.9 | 27.5 / 86.2 | 34.1 / 83.6 | 31.8 / 83.4 |
| Gemma3-27B-it (Team et al., 2025) | 37.4 / 88.0 | 23.8 / 81.0 | 25.0 / 81.2 | 24.5 / 81.5 | 34.3 / 82.9 | 31.8 / 82.6 |
| NLLB-moe-54B (Team et al., 2022) | 35.7 / 86.3 | 21.8 / 81.7 | 23.6 / 83.7 | 22.8 / 82.1 | 25.4 / 76.9 | 24.4 / 77.7 |
| Qwen3-Next-80B-A3B (Team, 2025) | 34.5 / 86.6 | 22.9 / 83.8 | 23.9 / 83.9 | 24.2 / 84.3 | 30.5 / 81.5 | 29.6 / 81.6 |
| **Models based on Text & Synthetic Speech** | | | | | | |
| Baseline (Text only) | 39.7 / 88.3 | 26.6 / 85.4 | 27.4 / 85.6 | 27.5 / 85.7 | 33.9 / 82.7 | 32.1 / 82.9 |
| SMT-9B | 40.4 / 89.5 | 27.3 / 86.9 | 28.3 / 87.1 | 28.3 / 87.4 | 33.4 / 83.0 | 32.2 / 83.4 |

Underlined denotes previous state-of-the-art models, while highlighted surpasses the previous models.

Table 4: Translation Performance on FLORES-200 and WMT24++ (spBLEU / COMET) MT Benchmarks. The average character length of the input English text is **130.4** for FLORES-200 and **191.3** for WMT24++. The notation $< 200$ indicates that the input English text length is within 200 characters. Detailed results are summarized in Tables 11, and 12 in the Appendix.

### 3.4.2 EXPERIMENTAL RESULTS FOR MACHINE TRANSLATION

**Language Support.** Our model exhibits strong language support, surpassing existing MMT models. Specifically, Table 4 details results for 108 translation directions on the **FLORES-200** benchmark, encompassing major source languages—English (eng), Japanese (jpn), Korean (kor), and Chinese (cmn)—to 27 target languages. Furthermore, we evaluate on the **WMT24++** benchmark for **en→22** directions. The complete list of supported languages is provided in Table 8 in the Appendix.

**Scalable Multilingualism.** The consistent performance gain underscores our method's advantages: scalability and multilingual capability. As shown in the Table 4, our model not only performs exceptionally well on the eng→xx task, but also delivers impressive gains on jpn→xx, kor→xx, and cmn→xx directions. The average spBLEU scores for these language groups are 27.3, 28.3, and 28.3 respectively, all of which are the highest in their respective categories.

**SMT in Low-Scoring Directions.** As shown in Figure 3, the **SMT-9B** model outperforms both the Baseline and DeepSeek models, particularly in low-resource translation directions like Khmer (khm), Lao (lao), and Burmese (mya), indicating its greater robustness in data-scarce language pairs. Beyond this, we note an underperforming high-resource language, Hindi (hin), whose translation metrics are lower than many low-resource counterparts.

**Translation Text Length.** As shown in Table 4, the WMT24++ dataset contains numerous extremely long texts, leading to noise (e.g., word omissions or duration exceeding $30s$) in the synthesized speech. Although the model's performance on the overall dataset is moderate, it exhibits good performance within the $< 200$ range. More importantly, the model's performance does not significantly degrade compared to the baseline, even when receiving noisy speech input, which fully demonstrates the model's robustness.

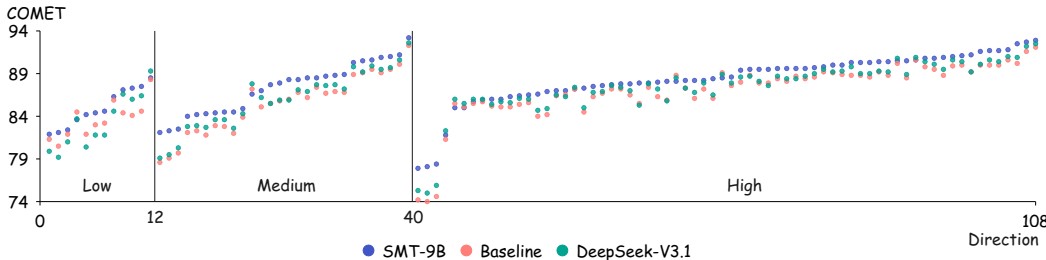

Figure 3: COMET Results by Resource Level, Categorized as Low, Medium, and High. Our model shows an improvement in translation scores, particularly for low-scoring translation directions.

| Input | | | eng → xx | | | | | | spBLEU / COMET ↑ |
|---|---|---|---|---|---|---|---|---|---|
| Text | AS | SS | ara | deu | fra | ind | jpn | tur | Avg. |
| ✓ | | | 37.7 / 86.3 | 45.2 / 88.0 | 32.1 / 86.9 | 47.9 / 91.5 | 31.5 / 90.7 | 36.7 / 88.8 | 38.5 / 88.7 |
| | ✓ | | 32.6 / 82.2 | 36.6 / 82.2 | 27.9 / 82.6 | 36.8 / 85.9 | 26.9 / 86.5 | 29.3 / 83.6 | 31.7 / 83.8 |
| | | ✓ | 34.1 / 83.5 | 39.0 / 84.0 | 28.9 / 83.8 | 36.9 / 87.4 | 27.1 / 87.4 | 30.3 / 85.0 | 32.7 / 85.4 |
| ✓ | ✓ | | 40.1 / 86.8 | 46.5 / 88.3 | 33.6 / 87.4 | 48.4 / 91.6 | 33.6 / 90.6 | 37.9 / 89.1 | 40.0 / 89.0 |
| ✓ | | ✓ | 40.1 / 86.8 | 46.5 / 88.2 | 33.6 / 87.4 | 48.5 / 91.6 | 33.5 / 90.7 | 37.8 / 89.1 | 40.0 / 89.0 |

Table 5: Ablation Study on the CoVoST-2 Benchmark. A comparison of configurations with different modality inputs. (AS denotes authentic speech; SS denotes synthetic speech)

| Models | eng → xx | | | | | | spBLEU / COMET ↑ |
|---|---|---|---|---|---|---|---|
| | jpn | cmn | tha | khm | lao | mya | Avg. |
| Baseline | 33.3 / 91.3 | 41.6 / 89.2 | 42.5 / 88.7 | 24.1 / 84.2 | 31.5 / 84.7 | 20.1 / 88.1 | 32.2 / 87.7 |
| SMT-9B | 35.2 / 92.7 | 42.6 / 91.2 | 44.1 / 90.3 | 25.6 / 83.6 | 34.2 / 86.3 | 24.3 / 88.5 | 34.3 / 88.8 |
| w/o SE | 34.8 / 92.1 | 42.3 / 89.3 | 42.5 / 89.7 | 23.0 / 81.7 | 31.7 / 84.3 | 23.4 / 86.8 | 33.0 / 87.3 |

Table 6: Ablation Study on Self-Evolution (SE) Mechanism on the FLORES-200 benchmark.

### 3.4.3 ABLATION STUDY

**Authentic Speech vs. Synthetic Speech.** As shown in Table 5, experimental results reveal that the difference between authentic and synthetic speech has minimal impact on multimodal machine translation performance. Surprisingly, synthetic speech achieves better S2TT performance, likely due to the absence of background noise. Experimental results demonstrate strong semantic consistency between authentic and synthetic speech.

**The Impact of the Self-Evolution Mechanism.** As shown in Table 6, we found that after MLLM pre-training, the model's performance on high-resource languages improved. However, due to the imbalance of multilingual data, the performance on low-resource languages like Khmer (khm), Lao (lao), and Burmese (mya) actually decreased on the COMET metric. Therefore, we introduced the self-evolution mechanism to enhance the model's performance on these low-resource directions.

**Self-Evolution Rounds on Low-Resource Languages.** Figure 4 shows the improvements from self-evolution for low-resource languages, with round 3 achieving best average gains of +1.9, +2.0, and +1.7 COMET on khm, lao, and mya, respectively. We observe that the first round yields the most significant improvement, later rounds give fewer benefits. The average improvement peaks at round 3 and then remains stable.

**Human Evaluation for MT and SMT.** Manual review of evaluation samples revealed that the performance gain from adding the speech modality is likely due to a reduction in **under-translation**, which decreased from 5.2% to 3.5%, as shown in Figure 5. The introduction of the speech modality provides prosodic cues as additional signals that effectively help correct the attention weighting, thereby mitigating this problem.

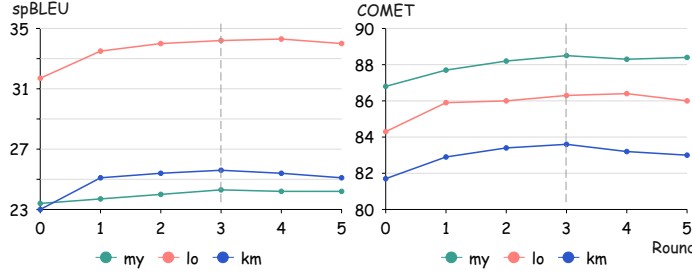

Figure 4: Self-Evolution Rounds of spBLEU / COMET (eng→xx) on FLORES-200 benchmark.

| Case | Translation from English to Chinese, Japanese, and Spanish |
|------|------------------------------------------------------------|
| - Input
- Ground Truth | Singapore is generally an extremely safe place to be and very easy to navigate, and you can buy almost anything after arriving.
通常来讲，新加坡是一个非常安全的地方，导航也很容易。到达后你几乎可以买到任何东西。 |
| Baseline
SMT-9B | 新加坡总体来说是一个非常安全的地方，而且很容易导航，你几乎可以买到任何东西。
新加坡通常是一个非常安全的地方，而且很容易导航，抵达后几乎可以买到任何东西。 |
| - Input
- Ground Truth | The patient had been to Nigeria, where some cases of the Ebola virus have occurred.
この患者は、エボラウイルスの症例がいくつか発生しているナイジェリアに行っていた。 |
| Baseline
SMT-9B | 患者は、エボラウイルスが発生したナイジェリアにいた。
患者はナイジェリアに滞在していたが、ナイジェリアではエボラウイルス感染例が報告されている。 |
| - Input
- Ground Truth | Workers must often get their superiors' approval for any decisions they make, and are expected to obey their superiors' instructions without question.
Con frecuencia, los trabajadores deben contar con la aprobación de sus superiores para la toma de decisiones y se espera que obedezcan sus instrucciones sin cuestionamiento. |
| Baseline
SMT-9B | Los trabajadores deben obtener la aprobación de sus superiores para cualquier decisión que tomen y se espera que obedezcan las instrucciones de sus superiores sin cuestionarlas.
Con frecuencia, los trabajadores deben obtener la aprobación de sus superiores para cualquier decisión que tomen y se espera que obedezcan las instrucciones de sus superiores sin cuestionarlas. |

Figure 5: Case Study for Under-Translation. Having undergone speech pre-training, MLLMs align text words with speech. The SMT model, which receives this speech-text fusion input, is prevented from ignoring the input text, thereby mitigating omission errors.

## 4 RELATED WORK

**Multimodal Machine Translation.** MMT research has primarily followed two distinct paths: image-based and image-free approaches. Image-based methods, exemplified by foundational work on the Multi30K dataset (Elliott et al., 2016), utilize paired visual and textual data to improve translation quality. In contrast, image-free approaches emerged to tackle the challenges of data scarcity. These methods employ various techniques, such as target-end retrieval (Hitschler et al., 2016), multi-task learning (Elliott & Kádár, 2017), and even visual generation using advanced models like GANs and diffusion models (Rombach et al., 2022), to generate or retrieve supplementary information without relying on a pre-existing image dataset.

**Multimodal Large Language Model.** MLLMs (Chen et al., 2024b; Du et al., 2025c;b) typically feature three core components: an LLM backbone, a modality encoder, and a modality adapter. Our framework specifically leverages this architecture to handle both speech and text. The speech encoder, inspired by models like Whisper (Radford et al., 2023), is responsible for extracting rich speech features from the audio input. Following this, the speech adapter (Li et al., 2023b) projects these features into the same hidden dimension as the LLM, enabling seamless integration. The processed speech features are then concatenated with the original text embeddings. This unified representation is fed into the LLM backbone, which processes both modalities jointly to generate the final translated text.

**Self-Evolution.** The concept of self-evolution (Liu et al., 2021) empowers models to autonomously acquire, refine, and learn from self-generated experiences. As outlined in recent surveys (Tao et al., 2024), this process typically involves a four-phase iterative cycle: (1) experience acquisition, (2) experience refinement, (3) updating, and (4) evaluation. Each iteration is designed to achieve a specific evolutionary objective. In our implementation, the process begins with the experience acquisition phase, where we generate synthetic speech data. This is followed by a refinement phase that involves the annotation of positive and negative samples. This newly labeled data is then used to update the model, which is subsequently evaluated for its machine translation performance.

## 5 CONCLUSION

In this paper, we present the Speech-guided Machine Translation (SMT) framework, a novel approach that overcomes the limitations of traditional image-based multimodal translation. Our framework integrates a TTS model with an MLLM, leveraging speech as a complementary modality to text. A key feature is the Self-Evolution Mechanism, which autonomously generates and refines training data. This significantly reduces the need for human-annotated data in low-resource languages, making the system more scalable and practical. Our experiments show that SMT-9B achieves SOTA performance on benchmarks such as Multi30K and FLORES-200.

## 6    LIMITATION

Unlike image-based methods, our speech-guided machine translation approach can cover a broader range of languages. However, we are still limited by the languages supported by the TTS models, as we need to synthesize speech from text. Although recent advancements in TTS technology have enabled the synthesis of dozens of languages, open-source TTS models still have limited language coverage.

## 7    THE USE OF LARGE LANGUAGE MODELS

In this paper, LLMs are not used for ideation but are utilized for checking grammatical rules.

## 8    REPRODUCIBILITY STATEMENT

All models and datasets tested in this research are open-source.

## ACKNOWLEDGEMENTS

The research in this article is supported by the National Science and Technology Major Program (Grant No. 2024ZD01NL00101), the National Science Foundation of China (U22B2059, 62276083, 62506182), National Key Research and Development Program of China (2025YFE0200500), the Key Research and Development Program of Heilongjiang Province (2022ZX01A28) and the 5G Application Innovation Joint Research Institute's Project (A003), and the Major Key Project of PCL.

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

# A  APPENDIX

| ISO-3 | Language | Script | Family | Subgrouping | Resource |
|-------|----------|--------|--------|-------------|----------|
| ben | Bemba | Latin | Atlantic-Congo | Benue-Congo | Low |
| ces | Czech | Latin | Indo-European | Balto-Slavic | High |
| cmn | Chinese | Han | Sino-Tibetan | Sinitic | High |
| deu | German | Latin | Indo-European | Germanic | High |
| eng | English | Latin | Indo-European | Germanic | High |
| fra | French | Latin | Indo-European | Italic | High |
| hau | Hausa | Latin | Afro-Asiatic | Chadic | Low |
| hin | Hindi | Devanagari | Indo-European | Indo-Aryan | High |
| lav | Latvian | Latin | Indo-European | Balto-Slavic | High |
| spa | Spanish | Latin | Indo-European | Italic | High |
| tur | Turkish | Latin | Turkic | Common Turkic | High |

Table 7: 11 Languages Supported by Image-Guided MMT datasets. The resource of each language is determined according to the taxonomy classes by (Joshi et al., 2020).

| ISO-3 | Language | Script | Family | Subgrouping | Resource |
|-------|----------|--------|--------|-------------|----------|
| ara | Arabic | Arabic | Afro-Asiatic | Semitic | High |
| ben | Bengali | Bengali | Indo-European | Indo-Aryan | Med |
| ces | Czech | Latin | Indo-European | Balto-Slavic | High |
| cmn | Chinese | Han | Sino-Tibetan | Sinitic | High |
| deu | German | Latin | Indo-European | Germanic | High |
| eng | English | Latin | Indo-European | Germanic | High |
| fas | Persian | Arabic | Indo-European | Iranian | High |
| fra | French | Latin | Indo-European | Italic | High |
| heb | Hebrew | Hebrew | Afro-Asiatic | Semitic | Med |
| hin | Hindi | Devanagari | Indo-European | Indo-Aryan | High |
| ind | Indonesian | Latin | Austronesian | Malayo-Polynesian | Med |
| ita | Italian | Latin | Indo-European | Italic | High |
| jpn | Japanese | Japanese | Japonic | Japanesic | High |
| khm | Khmer | Khmer | Austroasiatic | Khmeric | Low |
| kor | Korean | Hangul | Koreanic | Korean | High |
| lao | Lao | Lao | Tai-Kadai | Kam-Tai | Low |
| msa | Malay | Latin | Austronesian | Malayo-Polynesian | Med |
| mya | Burmese | Myanmar | Sino-Tibetan | Burmo-Qiangic | Low |
| nld | Dutch | Latin | Indo-European | Germanic | High |
| pol | Polish | Latin | Indo-European | Balto-Slavic | High |
| por | Portuguese | Latin | Indo-European | Italic | High |
| rus | Russian | Cyrillic | Indo-European | Balto-Slavic | High |
| spa | Spanish | Latin | Indo-European | Italic | High |
| tgl | Tagalog | Latin | Austronesian | Malayo-Polynesian | Med |
| tha | Thai | Thai | Tai-Kadai | Kam-Tai | Med |
| tur | Turkish | Latin | Turkic | Common Turkic | High |
| urd | Urdu | Arabic | Indo-European | Indo-Aryan | Med |
| vie | Vietnamese | Latin | Austroasiatic | Vietic | High |

Table 8: 28 Languages Supported by Our Model. The resource of each language is determined according to the taxonomy classes by (Joshi et al., 2020).

| Model | Task | Description | Dataset | Split | Data Size | Metric |
|---|---|---|---|---|---|---|
| MLLM | ASR | Automatic Speech Recognition | FLEURS$^\dagger$ 
 Common Voice 19 | train 
 train | ∼160h 
 ∼3000h | WER ↓ |
| | SMT | Speech-Guided 
 Multimodal Machine Translation | FLEURS$^\dagger$ 
 Multi30K | train 
 train | ∼160h 
 ∼40h | spBLEU / COMET ↑ 
 BLEU / COMET ↑ |

Table 9: Summary of Training Datasets for SMT Models. Data size refers to the actual amount used for training, as we removed some overly long samples. † indicates that we performed data cleaning on the dataset. Since there is an overlap between the FLEURS and FLORES datasets, we removed the overlapping portions from the FLEURS training set.

| Task | Description | Dataset | Split | Metric |
|---|---|---|---|---|
| MT | Machine Translation | FLORES-200 
 WMT24++ | devtest 
 test | spBLEU / COMET ↑ |
| MMT | Multimodal Machine Translation | Multi30K | test | BLEU / COMET ↑ |
| S2TT | Speech-to-Text Translation | CoVoST-2 | test | spBLEU / COMET ↑ |

Table 10: Summary of Evaluation Benchmarks.

| Direction | DeepSeek -v3.1 | Gemma3 -27B | NLLB-moe -54B | Qwen3-Next -80B-A3B | Baseline | SMT -9B |
|---|---|---|---|---|---|---|
| eng → ara | 20.0 / 78.5 | 20.0 / 78.1 | 18.5 / 74.6 | 19.4 / 77.6 | 19.4 / 77.3 | 19.1 / 77.5 |
| eng → ben | 25.9 / 83.3 | 25.4 / 82.7 | 23.5 / 79.7 | 16.2 / 78.5 | 24.8 / 82.1 | 23.6 / 80.7 |
| eng → ces | 36.3 / 85.9 | 36.3 / 84.6 | 23.4 / 79.0 | 30.3 / 82.2 | 36.4 / 85.1 | 35.8 / 85.5 |
| eng → cmn | 34.3 / 84.9 | 36.4 / 83.5 | 18.0 / 69.5 | 37.4 / 84.9 | 36.9 / 83.8 | 36.5 / 85.3 |
| eng → deu | 37.9 / 82.6 | 37.9 / 81.9 | 28.5 / 76.3 | 36.2 / 81.7 | 37.7 / 82.3 | 37.3 / 82.3 |
| eng → fas | 29.9 / 83.1 | 32.9 / 83.1 | 25.8 / 78.0 | 27.7 / 80.7 | 32.1 / 83.1 | 31.9 / 83.8 |
| eng → fra | 48.1 / 82.7 | 47.4 / 82.2 | 34.8 / 75.5 | 45.0 / 82.0 | 44.3 / 82.2 | 45.0 / 81.9 |
| eng → heb | 37.4 / 82.6 | 36.6 / 82.3 | 33.9 / 79.4 | 26.8 / 76.7 | 38.8 / 83.5 | 38.3 / 84.5 |
| eng → hin | 19.6 / 74.0 | 19.5 / 73.4 | 16.0 / 65.5 | 12.6 / 68.5 | 19.3 / 71.0 | 19.6 / 70.2 |
| eng → ind | 38.2 / 86.8 | 37.6 / 86.3 | 30.6 / 80.8 | 36.6 / 86.0 | 37.3 / 85.3 | 37.2 / 86.0 |
| eng → ita | 45.2 / 84.7 | 46.2 / 84.4 | 33.3 / 78.6 | 41.9 / 83.7 | 45.0 / 84.6 | 44.2 / 85.2 |
| eng → jpn | 25.4 / 87.6 | 24.0 / 86.4 | 11.7 / 79.1 | 22.6 / 86.9 | 22.5 / 85.7 | 22.5 / 85.9 |
| eng → kor | 27.1 / 87.3 | 26.9 / 86.4 | 20.7 / 81.9 | 23.9 / 86.1 | 26.0 / 85.6 | 25.0 / 85.4 |
| eng → nld | 40.4 / 84.4 | 39.3 / 83.7 | 28.5 / 77.8 | 35.8 / 82.7 | 38.7 / 84.6 | 37.5 / 84.3 |
| eng → pol | 30.5 / 84.8 | 29.2 / 83.9 | 18.0 / 77.3 | 25.6 / 81.7 | 29.4 / 83.8 | 28.7 / 84.8 |
| eng → por | 40.7 / 83.4 | 40.0 / 82.9 | 28.7 / 77.2 | 38.6 / 82.7 | 39.5 / 83.0 | 39.3 / 83.4 |
| eng → rus | 29.6 / 83.4 | 31.4 / 82.7 | 23.2 / 76.6 | 28.8 / 81.9 | 29.2 / 81.9 | 29.9 / 83.5 |
| eng → spa | 48.4 / 83.7 | 48.7 / 83.6 | 36.0 / 77.7 | 46.2 / 83.0 | 48.5 / 83.7 | 46.1 / 83.8 |
| eng → tha | 32.6 / 85.1 | 33.8 / 84.8 | 22.3 / 77.9 | 29.6 / 83.5 | 32.4 / 82.8 | 31.7 / 83.7 |
| eng → tra | 36.0 / 85.5 | 36.6 / 84.3 | 27.0 / 79.0 | 30.6 / 83.0 | 36.6 / 84.6 | 36.3 / 84.2 |
| eng → urd | 30.5 / 79.8 | 30.3 / 79.0 | 29.0 / 73.7 | 23.5 / 75.8 | 33.3 / 79.8 | 32.4 / 80.5 |
| eng → vie | 36.6 / 84.8 | 37.2 / 84.1 | 26.5 / 77.7 | 35.9 / 83.9 | 37.6 / 83.7 | 37.1 / 84.2 |
| Avg. | 34.1 / 83.6 | 34.3 / 82.9 | 25.4 / 76.9 | 30.5 / 81.5 | 33.9 / 82.7 | 33.4 / 83.0 |

Table 11: spBLEU / COMET Scores on the WMT24++ Benchmark.

| Direction | DeepSeek -v3.1 | Gemma3 -27B | NLLB-moe -54B | Qwen3-Next -80B-A3B | Baseline | SMT -9B |
|---|---|---|---|---|---|---|
| eng → ara | 41.6 / 88.1 | 41.7 / 87.8 | 41.8 / 86.8 | 38.3 / 87.1 | 42.9 / 87.8 | 42.6 / 89.5 |
| eng → ben | 33.6 / 87.8 | 30.0 / 86.6 | 34.5 / 86.4 | 28.0 / 85.6 | 34.8 / 86.4 | 34.3 / 86.6 |
| eng → ces | 44.0 / 92.5 | 41.4 / 91.3 | 40.3 / 90.9 | 37.9 / 90.2 | 42.7 / 91.6 | 43.1 / 92.9 |
| eng → cmn | 35.7 / 89.2 | 37.2 / 88.8 | 22.4 / 78.0 | 37.0 / 89.2 | 41.6 / 89.2 | 42.6 / 91.2 |
| eng → deu | 48.5 / 89.0 | 46.9 / 88.7 | 43.8 / 87.1 | 46.2 / 88.5 | 47.1 / 88.5 | 47.8 / 89.7 |
| eng → fas | 35.1 / 89.0 | 35.3 / 88.7 | 34.4 / 87.2 | 30.9 / 86.8 | 38.7 / 88.9 | 38.3 / 90.3 |
| eng → fra | 56.3 / 89.2 | 55.6 / 88.8 | 54.6 / 87.7 | 55.1 / 88.8 | 57.7 / 89.1 | 57.1 / 90.0 |
| eng → heb | 47.8 / 89.7 | 45.4 / 89.1 | 45.0 / 88.4 | 33.1 / 83.4 | 46.3 / 89.3 | 46.8 / 91.0 |
| eng → hin | 37.9 / 82.3 | 36.8 / 81.7 | 38.6 / 80.7 | 31.9 / 79.9 | 41.3 / 81.1 | 41.0 / 81.8 |
| eng → ind | 50.0 / 92.6 | 49.5 / 92.0 | 48.1 / 91.1 | 48.7 / 92.1 | 52.6 / 92.2 | 52.4 / 93.2 |
| eng → ita | 39.1 / 89.3 | 39.1 / 89.4 | 37.1 / 88.1 | 37.7 / 89.0 | 38.8 / 89.3 | 39.4 / 90.4 |
| eng → jpn | 33.9 / 92.2 | 32.6 / 91.8 | 18.8 / 88.1 | 29.0 / 91.7 | 33.3 / 91.3 | 35.2 / 92.7 |
| eng → khm | 23.8 / 83.7 | 17.7 / 81.3 | 22.0 / 79.5 | 15.0 / 76.3 | 24.1 / 84.2 | 25.6 / 83.6 |
| eng → kor | 29.5 / 90.8 | 28.8 / 90.3 | 25.4 / 89.0 | 26.2 / 90.0 | 30.4 / 90.1 | 30.1 / 90.5 |
| eng → lao | 30.0 / 84.6 | 27.7 / 83.1 | 29.1 / 83.4 | 17.0 / 73.8 | 31.5 / 84.7 | 34.2 / 86.3 |
| eng → msa | 45.2 / 90.6 | 37.6 / 86.8 | 44.4 / 88.7 | 39.7 / 89.7 | 47.0 / 90.9 | 47.4 / 91.2 |
| eng → mya | 24.0 / 89.3 | 15.2 / 85.7 | 16.1 / 83.7 | 14.7 / 82.2 | 20.1 / 87.2 | 24.3 / 88.5 |
| eng → nld | 36.6 / 88.7 | 35.4 / 88.5 | 34.6 / 87.3 | 33.9 / 87.9 | 37.5 / 88.8 | 37.2 / 89.5 |
| eng → pol | 35.3 / 90.6 | 34.0 / 90.2 | 30.9 / 88.6 | 30.8 / 88.7 | 33.5 / 89.9 | 34.1 / 91.7 |
| eng → por | 55.3 / 90.4 | 55.3 / 90.4 | 51.0 / 88.8 | 53.9 / 90.1 | 53.2 / 90.0 | 55.4 / 91.1 |
| eng → rus | 43.0 / 90.9 | 41.2 / 90.1 | 38.8 / 88.8 | 40.2 / 90.1 | 41.4 / 90.1 | 41.6 / 92.5 |
| eng → spa | 34.4 / 87.3 | 33.9 / 87.2 | 32.3 / 85.9 | 33.4 / 87.0 | 35.5 / 87.2 | 36.5 / 88.2 |
| eng → tgl | 39.5 / 86.2 | 38.9 / 85.9 | 37.4 / 84.5 | 30.1 / 82.5 | 38.2 / 84.7 | 41.0 / 87.0 |
| eng → tha | 44.8 / 89.8 | 42.8 / 89.4 | 32.1 / 83.7 | 40.8 / 88.8 | 42.5 / 88.7 | 44.1 / 90.3 |
| eng → tur | 42.1 / 90.9 | 40.2 / 90.5 | 39.5 / 89.2 | 35.5 / 89.3 | 42.2 / 90.6 | 41.7 / 90.6 |
| eng → urd | 30.3 / 84.3 | 27.6 / 83.0 | 28.8 / 81.0 | 23.8 / 80.6 | 30.9 / 83.9 | 30.7 / 84.9 |
| eng → vie | 44.4 / 90.4 | 43.6 / 90.0 | 42.5 / 87.9 | 42.8 / 89.8 | 46.7 / 90.0 | 46.4 / 91.7 |
| jpn → ara | 27.3 / 84.7 | 26.8 / 84.1 | 23.3 / 80.8 | 24.1 / 83.6 | 26.5 / 84.0 | 26.8 / 86.6 |
| jpn → ben | 23.7 / 82.7 | 0.6 / 49.0 | 20.9 / 79.4 | 19.7 / 80.4 | 24.1 / 81.8 | 24.0 / 84.3 |
| jpn → ces | 26.7 / 90.4 | 25.9 / 89.1 | 20.2 / 86.1 | 23.2 / 89.1 | 25.8 / 89.8 | 27.2 / 90.8 |
| jpn → cmn | 27.5 / 88.4 | 27.9 / 88.2 | 15.1 / 74.8 | 27.2 / 88.3 | 32.0 / 88.7 | 33.3 / 89.6 |
| jpn → deu | 29.2 / 85.7 | 28.7 / 85.3 | 23.9 / 81.9 | 27.1 / 85.0 | 28.2 / 85.1 | 28.8 / 86.0 |
| jpn → eng | 32.7 / 88.5 | 33.4 / 88.5 | 33.2 / 87.4 | 32.4 / 88.5 | 36.9 / 88.8 | 37.8 / 88.1 |
| jpn → fas | 22.7 / 85.3 | 15.3 / 67.6 | 18.5 / 79.9 | 20.4 / 83.9 | 24.4 / 85.5 | 25.4 / 87.9 |
| jpn → fra | 34.0 / 86.0 | 34.0 / 85.8 | 29.8 / 83.0 | 32.2 / 85.4 | 32.8 / 85.7 | 34.7 / 85.9 |
| jpn → heb | 27.4 / 85.5 | 27.0 / 85.4 | 21.0 / 79.2 | 19.9 / 80.4 | 27.1 / 85.5 | 27.9 / 87.7 |
| jpn → hin | 24.2 / 75.3 | 24.0 / 74.8 | 21.0 / 71.4 | 19.8 / 73.2 | 24.1 / 74.2 | 24.7 / 77.9 |
| jpn → ind | 28.1 / 89.2 | 28.4 / 88.4 | 25.3 / 87.0 | 27.1 / 88.8 | 30.4 / 89.1 | 30.9 / 90.5 |
| jpn → ita | 27.3 / 87.4 | 26.9 / 87.2 | 22.1 / 84.0 | 25.1 / 86.9 | 26.3 / 87.2 | 26.7 / 87.8 |
| jpn → khm | 18.5 / 79.9 | 13.7 / 77.1 | 16.4 / 77.7 | 12.0 / 73.8 | 21.1 / 81.3 | 19.1 / 81.9 |
| jpn → kor | 23.6 / 88.7 | 23.5 / 88.2 | 19.8 / 84.8 | 21.6 / 88.5 | 23.5 / 88.4 | 23.5 / 89.6 |
| jpn → lao | 19.8 / 80.4 | 20.0 / 79.6 | 21.9 / 80.2 | 12.0 / 70.6 | 24.7 / 81.9 | 24.2 / 84.2 |
| jpn → msa | 25.9 / 87.1 | 21.9 / 84.0 | 23.2 / 84.7 | 21.4 / 86.4 | 27.3 / 86.8 | 28.6 / 88.3 |
| jpn → mya | 17.3 / 86.0 | 0.3 / 27.8 | 12.1 / 81.2 | 11.7 / 78.9 | 16.5 / 84.1 | 19.3 / 87.3 |
| jpn → nld | 24.7 / 86.3 | 24.9 / 86.3 | 20.2 / 82.1 | 23.2 / 85.8 | 25.8 / 86.5 | 25.9 / 87.0 |
| jpn → pol | 24.6 / 89.5 | 25.1 / 89.3 | 18.9 / 84.7 | 21.8 / 88.2 | 23.9 / 89.2 | 24.2 / 89.8 |
| jpn → por | 30.4 / 87.4 | 31.4 / 87.3 | 26.9 / 84.6 | 29.4 / 87.1 | 31.3 / 87.2 | 32.3 / 87.2 |
| jpn → rus | 28.1 / 89.0 | 27.1 / 87.9 | 23.8 / 86.0 | 26.2 / 88.4 | 26.9 / 88.6 | 27.4 / 90.3 |
| jpn → spa | 24.8 / 86.0 | 23.8 / 85.5 | 20.9 / 83.5 | 22.9 / 85.5 | 24.3 / 85.5 | 25.3 / 85.0 |
| jpn → tgl | 24.1 / 82.8 | 23.4 / 82.0 | 18.8 / 78.7 | 17.8 / 80.0 | 23.1 / 82.1 | 24.9 / 84.0 |
| jpn → tha | 35.5 / 86.9 | 35.1 / 86.8 | 24.0 / 79.9 | 32.8 / 86.3 | 33.9 / 86.2 | 34.8 / 88.5 |
| jpn → tur | 25.9 / 87.0 | 25.6 / 86.7 | 21.4 / 82.2 | 21.9 / 85.5 | 26.1 / 86.5 | 26.4 / 87.8 |
| jpn → urd | 20.6 / 79.1 | 18.7 / 78.1 | 19.3 / 75.9 | 15.6 / 76.1 | 20.1 / 78.6 | 20.8 / 82.1 |
| jpn → vie | 29.7 / 88.4 | 30.4 / 88.3 | 26.7 / 85.5 | 28.8 / 88.0 | 31.4 / 88.1 | 32.2 / 89.6 |
| kor → ara | 28.4 / 84.9 | 28.5 / 85.0 | 25.9 / 83.6 | 25.3 / 84.1 | 26.9 / 84.2 | 27.4 / 86.9 |
| kor → ben | 25.2 / 82.9 | 0.8 / 30.2 | 22.7 / 81.2 | 20.7 / 80.6 | 24.9 / 82.3 | 24.7 / 84.2 |
| kor → ces | 29.0 / 90.1 | 27.1 / 89.3 | 23.4 / 87.9 | 23.5 / 88.5 | 27.2 / 89.5 | 27.8 / 90.8 |
| kor → cmn | 28.4 / 87.6 | 29.3 / 87.4 | 19.1 / 80.4 | 28.5 / 87.6 | 32.4 / 87.9 | 33.8 / 89.5 |
| kor → deu | 30.6 / 85.6 | 29.3 / 84.7 | 25.5 / 83.2 | 27.7 / 84.9 | 29.1 / 85.1 | 29.8 / 86.3 |
| kor → eng | 35.4 / 88.9 | 35.9 / 88.9 | 34.3 / 87.9 | 34.9 / 88.8 | 39.1 / 89.1 | 40.3 / 88.5 |
| kor → fas | 24.5 / 85.8 | 24.0 / 84.0 | 22.3 / 84.2 | 21.4 / 84.1 | 25.6 / 85.9 | 26.2 / 88.1 |
| kor → fra | 35.3 / 85.4 | 35.4 / 85.2 | 31.9 / 83.6 | 33.7 / 84.9 | 33.6 / 85.2 | 35.9 / 86.0 |
| kor → heb | 28.4 / 85.9 | 29.1 / 85.8 | 25.6 / 84.4 | 20.2 / 80.7 | 28.1 / 85.8 | 28.8 / 87.9 |
| kor → hin | 25.2 / 75.0 | 25.3 / 74.4 | 23.5 / 72.5 | 21.1 / 73.2 | 24.9 / 74.0 | 25.7 / 78.1 |
| kor → ind | 30.7 / 89.9 | 31.0 / 89.7 | 27.4 / 88.5 | 28.9 / 89.4 | 31.7 / 89.5 | 32.0 / 90.6 |
| kor → ita | 27.7 / 86.9 | 24.2 / 82.3 | 23.4 / 84.9 | 25.5 / 86.4 | 26.8 / 86.7 | 26.9 / 87.6 |
| kor → jpn | 26.7 / 90.1 | 27.8 / 90.4 | 15.6 / 86.8 | 25.1 / 90.1 | 26.9 / 90.0 | 28.8 / 91.6 |
| kor → khm | 19.3 / 79.2 | 12.7 / 75.1 | 17.5 / 79.1 | 12.1 / 72.9 | 21.3 / 80.5 | 20.0 / 82.1 |
| kor → lao | 22.7 / 81.8 | 21.0 / 80.3 | 22.1 / 81.9 | 12.5 / 72.3 | 25.7 / 83.0 | 24.9 / 84.4 |
| kor → msa | 27.4 / 87.7 | 22.6 / 84.6 | 24.2 / 86.0 | 22.7 / 86.8 | 28.5 / 87.4 | 29.4 / 88.5 |
| kor → mya | 19.5 / 86.6 | 1.2 / 38.1 | 12.1 / 82.9 | 12.1 / 80.0 | 15.8 / 84.4 | 19.2 / 87.1 |
| kor → nld | 26.0 / 86.5 | 25.7 / 85.7 | 22.9 / 84.6 | 23.8 / 85.5 | 26.0 / 86.4 | 26.3 / 87.0 |
| kor → pol | 25.8 / 89.3 | 25.5 / 88.9 | 21.5 / 86.9 | 22.6 / 87.8 | 24.4 / 88.9 | 24.6 / 90.0 |
| kor → por | 32.7 / 87.6 | 28.4 / 85.5 | 27.9 / 85.4 | 31.3 / 87.2 | 32.2 / 87.2 | 33.6 / 87.7 |
| kor → rus | 29.5 / 88.9 | 28.3 / 88.1 | 25.5 / 87.3 | 26.7 / 88.3 | 27.5 / 88.5 | 28.2 / 90.5 |
| kor → spa | 24.8 / 85.5 | 24.1 / 85.2 | 21.3 / 83.6 | 23.0 / 85.0 | 24.6 / 85.1 | 25.6 / 85.0 |
| kor → tgl | 25.1 / 83.6 | 24.9 / 83.4 | 21.2 / 81.3 | 18.8 / 80.8 | 23.8 / 82.9 | 25.8 / 84.4 |
| kor → tha | 36.6 / 87.6 | 35.5 / 87.5 | 26.2 / 83.3 | 33.4 / 86.8 | 33.5 / 86.7 | 35.0 / 88.7 |
| kor → tur | 28.4 / 87.2 | 27.4 / 86.8 | 24.0 / 84.8 | 23.6 / 85.4 | 27.2 / 86.3 | 27.9 / 88.0 |
| kor → urd | 21.4 / 79.5 | 19.6 / 78.6 | 20.3 / 77.2 | 16.3 / 76.2 | 21.0 / 79.1 | 21.4 / 82.3 |
| kor → vie | 32.4 / 88.7 | 31.8 / 88.3 | 29.1 / 87.0 | 29.8 / 88.0 | 32.3 / 88.4 | 33.1 / 89.6 |
| cmn → ara | 28.4 / 85.0 | 28.0 / 84.8 | 25.4 / 82.5 | 26.0 / 84.2 | 27.6 / 84.5 | 28.0 / 87.3 |
| cmn → ben | 24.7 / 83.6 | 10.2 / 62.2 | 21.2 / 79.9 | 20.6 / 81.8 | 24.4 / 82.8 | 24.3 / 84.5 |
| cmn → ces | 29.1 / 90.6 | 27.7 / 89.5 | 21.7 / 85.2 | 25.7 / 89.1 | 28.0 / 89.9 | 28.9 / 91.0 |
| cmn → deu | 30.7 / 86.1 | 29.2 / 85.5 | 24.2 / 81.6 | 28.6 / 85.6 | 29.5 / 85.4 | 30.0 / 86.4 |
| cmn → eng | 35.0 / 87.9 | 35.0 / 87.6 | 35.0 / 86.6 | 35.5 / 87.8 | 36.1 / 87.6 | 39.2 / 88.6 |
| cmn → fas | 24.9 / 86.5 | 17.3 / 72.4 | 20.5 / 81.7 | 22.2 / 84.8 | 26.4 / 86.1 | 27.0 / 88.4 |
| cmn → fra | 36.1 / 86.0 | 35.7 / 85.6 | 31.4 / 82.2 | 34.7 / 85.6 | 34.6 / 85.6 | 36.9 / 86.5 |
| cmn → heb | 27.9 / 85.9 | 27.3 / 85.5 | 24.1 / 82.6 | 21.0 / 80.9 | 28.8 / 86.0 | 29.4 / 88.3 |
| cmn → hin | 25.2 / 75.9 | 24.8 / 75.4 | 21.9 / 71.6 | 21.4 / 73.9 | 25.3 / 74.6 | 26.4 / 78.4 |
| cmn → ind | 30.5 / 89.5 | 30.0 / 88.7 | 27.0 / 87.0 | 29.5 / 89.2 | 32.2 / 89.1 | 32.3 / 90.9 |
| cmn → ita | 29.1 / 87.9 | 28.7 / 87.6 | 23.2 / 83.6 | 26.9 / 87.3 | 27.1 / 87.2 | 28.2 / 88.2 |
| cmn → jpn | 26.2 / 91.0 | 25.8 / 90.7 | 14.5 / 84.7 | 23.3 / 90.7 | 25.3 / 90.6 | 27.3 / 91.8 |
| cmn → khm | 19.7 / 81.0 | 13.9 / 77.3 | 16.1 / 77.9 | 12.4 / 74.2 | 22.0 / 81.9 | 20.2 / 82.4 |
| cmn → kor | 22.8 / 88.6 | 23.1 / 87.9 | 18.0 / 84.8 | 20.6 / 88.4 | 22.7 / 88.0 | 22.9 / 89.4 |
| cmn → lao | 22.2 / 81.8 | 21.0 / 80.9 | 21.7 / 79.7 | 12.6 / 71.6 | 26.2 / 83.2 | 25.4 / 84.6 |
| cmn → msa | 26.7 / 87.2 | 24.9 / 86.0 | 24.8 / 84.6 | 23.0 / 86.5 | 28.6 / 86.8 | 29.7 / 88.9 |
| cmn → mya | 18.7 / 86.4 | 0.2 / 27.1 | 13.3 / 79.2 | 12.0 / 79.5 | 16.5 / 84.6 | 19.5 / 87.5 |
| cmn → nld | 26.8 / 86.8 | 25.5 / 86.3 | 21.5 / 82.4 | 24.5 / 85.9 | 26.5 / 86.3 | 27.0 / 87.5 |
| cmn → pol | 26.3 / 89.8 | 25.7 / 89.1 | 20.2 / 83.9 | 23.6 / 88.2 | 25.4 / 88.8 | 25.9 / 90.3 |
| cmn → por | 32.2 / 87.8 | 31.9 / 87.2 | 28.1 / 84.4 | 31.7 / 87.5 | 32.9 / 87.4 | 34.0 / 87.9 |
| cmn → rus | 31.2 / 89.5 | 28.7 / 88.2 | 24.8 / 85.5 | 28.1 / 88.9 | 28.7 / 88.8 | 28.9 / 90.9 |
| cmn → spa | 25.4 / 86.0 | 24.8 / 85.7 | 21.9 / 82.9 | 24.6 / 85.7 | 25.1 / 85.5 | 26.6 / 85.7 |
| cmn → tgl | 24.6 / 82.6 | 24.6 / 82.3 | 20.3 / 78.1 | 18.4 / 79.7 | 24.3 / 82.0 | 26.0 / 84.5 |
| cmn → tha | 37.3 / 87.7 | 20.4 / 73.1 | 27.1 / 82.6 | 34.2 / 87.1 | 35.5 / 86.9 | 36.2 / 88.8 |
| cmn → tur | 26.8 / 86.8 | 26.0 / 86.3 | 21.5 / 82.0 | 24.0 / 85.4 | 26.5 / 86.1 | 27.3 / 88.2 |
| cmn → urd | 21.8 / 80.3 | 19.7 / 79.3 | 17.4 / 74.0 | 16.4 / 76.8 | 21.5 / 79.7 | 21.6 / 82.5 |
| cmn → vie | 32.7 / 89.2 | 32.4 / 88.9 | 30.0 / 86.5 | 31.6 / 88.8 | 34.0 / 88.8 | 34.6 / 90.4 |
| Avg. | 30.1 / 86.7 | 27.7 / 83.0 | 26.0 / 83.5 | 26.4 / 84.7 | 30.3 / 86.2 | 31.1 / 87.7 |

Table 12: spBLEU / COMET Scores on the FLORES-200 Benchmark.

