# OpenReview forum: "Scalable Multilingual Multimodal Machine Translation with Speech-Text Fusion"
_ICLR.cc/2026/Conference — ICLR 2026 Poster_

### Official Review · Reviewer_wxJV · 2025-10-29

**Soundness:** 3
**Presentation:** 2
**Contribution:** 3
**Rating:** 4
**Confidence:** 2

**Summary:**

The authors propose a Speech-guided Multimodal Machine Translation (SMMT) framework that integrates both speech and text as fused inputs into a multimodal large language model (MLLM) to enhance translation quality. The system comprises a multimodal LLM and a text-to-speech (TTS) module, augmented by a self-evolution mechanism that enables iterative model refinement. Experimental results demonstrate competitive performance on traditional machine translation benchmarks and promising improvements on the Multi30K multimodal dataset.

**Strengths:**

- Innovative self-evolution mechanism: The framework introduces a novel self-improvement process that leverages translation performance metrics as optimization objectives, facilitating continuous enhancement through iterative evolution cycles.
- Comprehensive multimodal and multilingual support: The approach effectively unifies speech and text modalities and extends to multiple languages, showing potential for broader generalization.

**Weaknesses:**

- Limited evaluation scope: Experiments are primarily conducted on a small set of benchmarks (e.g., Multi30K, FLORES-200), leaving open questions about generalization to large-scale and domain-diverse datasets.
- Dependence on external components: The framework relies heavily on open-source models, such as the TTS model, which may constrain performance consistency and scalability.
- Lack of detailed ablations: It remains unclear how much each component (speech input, multimodal fusion, self-evolution) contributes to the observed performance gains.

**Questions:**

- Could the authors clarify the conceptual and methodological differences between the proposed self-evolution mechanism and reinforcement learning approaches typically used for LLM fine-tuning?

---

> ### Author Response · Authors · 2025-11-21
> **(1/2) Response to Reviewer wxJV**
>
> ### **Thanks for the insightful comments. We mainly have updated Table 3, 4, 9, 12, 13 and Figure 5 in the revised PDF version.**
>
> > **[W1]** **Limited evaluation scope: Experiments are primarily conducted on a small set of benchmarks (e.g., Multi30K, FLORES-200), leaving open questions about generalization to large-scale and domain-diverse datasets.**
> >
>
> We conducted experiments on the **Multi30k**, **FLORES**, and **WMT24++** datasets. Multi30k is the most common multimodal machine translation dataset. Since we need to evaluate 28 languages, we select the most widely used multilingual machine translation datasets, such as FLORES and WMT.
>
> As shown in **Table 3** and **Table 4**, we have **added** experiments for the **eng $\rightarrow$ ces** direction on the **Multi30K** dataset and for the **eng $\rightarrow$ 22** directions on the **WMT24++** dataset. Experimental results show that our model, with only 10 billion (10B) parameters, surpassed models with comparable parameter sizes across all three datasets, even exceeding the performance of Deepseek-V3.1-671B (input text length<200).
>
> Averaging Text Length:  Multi30K (59.3), Flores-200 (130.4), WMT24++ (191.3).
>
> | **Multi30K（eng → xx）** | deu  | deu | deu  | fra  | fra  | fra  | **ces** | **ces** |
> | --- | --- | --- | --- | --- | --- | --- | --- | --- |
> | **BLEU / COMET** | Test2016 | Test2017 | MSCOCO | Test2016 | Test2017 | MSCOCO | **Test2016** | **Test2018** |
> | DeepSeek-V3.1-671B | 44.2 / 87.3 | 41.1 / 86.8 | 36.4 / 83.2 | 55.3 / 88.2 | 54.0 / 87.7 | 53.5 / 85.8 | 37.9 / 90.7 | 35.9 / 89.7 |
> | Gemma3-27B-it | 43.7 / 87.1 | 40.3 / 86.3 | 36.1 / 83.2 | 55.4 / 87.9 | 54.3 / 87.9 | 49.6 / 85.0 | 36.4 / 89.9 | 35.9 / 89.1 |
> | NLLB-moe-54B | 41.4 / 86.2 | 39.7 / 85.8 | 34.7 / 82.1 | 55.1 / 87.4 | 54.8 / 87.7 | 53.3 / 85.3 | 35.7 / 88.9 | 35.8 / 88.3 |
> | Qwen3-Next-80B-A3B | 41.6 / 86.3 | 37.6 / 85.9 | 31.9 / 82.5 | 53.2 / 87.8 | 51.9 / 87.6 | 50.4 / 85.1 | 29.2 / 87.2 | 27.9 / 85.9 |
> | Soul-Mix  | 44.2 / - | 37.1 / - | 34.2 / - | 64.7 / - | 57.4 / - | 49.2 / - | 36.5 / - | 32.8 / - |
> | ConsQA-MMT  | 44.2 / - | 37.6 / - | 34.3 / - | 64.8 / - | 58.3 / - | 48.5 / - | 34.7 / - | 30.3 / - |
> | Bridge  | 42.5 / - | 36.0 / - | 32.0 / - | 63.7 / - | 56.2 / - | 46.3 / - | 35.2 / - | 31.2 / - |
> | IMAGE  | 45.3 / 83.1 | 38.6 / 81.9 | 37.5 / 78.8 | 67.5 / 88.3 | 61.5 / 86.6 | 49.3 / 82.5 | - | - |
> | Baseline (text-only) | 42.9 / 87.0 | 38.8 / 86.4 | 34.3 / 82.7 | 52.4 / 87.7 | 52.0 / 87.9 | 52.6 / 86.1 | 34.1 / 89.9 | 34.8 / 89.0 |
> | Baseline + Lora (text-only) | 44.0 / 87.0 | 39.4 / 86.4 | 35.3 / 83.0 | 55.5 / 88.1 | 54.0 / 88.2 | 53.4 / 85.9 | 37.2 / 90.0 | 35.7 / 89.1 |
> | SMMT-10B (Ours) | **47.0** / **88.6** | **41.8** / **88.1** | **38.5** / **84.5** | **67.0** / **91.0** | **62.1** / **90.7** | **55.3** / **87.3** | **41.4** / **91.7** | **39.9** / **90.7** |
>
> | **spBLEU / COMET** | FLORES-200 (eng → 27) | FLORES-200 (jpn → 27) | FLORES-200 (kor → 27) | FLORES-200 (cmn → 27) | **WMT24++ (eng → 22)** | **WMT24++ (eng → 22 (<200))** |
> | --- | --- | --- | --- | --- | --- | --- |
> | DeepSeek-V3.1-671B | 39.3 / 88.9 | 26.1 / 85.7 | 27.7 / 85.9 | 27.5 / 86.2 | 34.1 / **83.6** | 31.8 / 83.4 |
> | Gemma3-27B-it | 37.4 / 88.0 | 23.8 / 81.0 | 25.0 / 81.2 | 24.5 / 81.5 | **34.3** / 82.9 | 31.8 / 82.6 |
> | NLLB-moe-54B | 35.7 / 86.3 | 21.8 / 81.7 | 23.6 / 83.7 | 22.8 / 82.1 | 25.4 / 76.9 | 24.4 / 77.7 |
> | Qwen3-Next-80B-A3B | 34.5 / 86.6 | 22.9 / 83.8 | 23.9 / 83.9 | 24.2 / 84.3 | 30.5 / 81.5 | 29.6 / 81.6 |
> | Baseline (text-only) | 39.7 / 88.3 | 26.6 / 85.4 | 27.4 / 85.6 | 27.5 / 85.7 | 33.9 / 82.7 | 32.1 / 82.9 |
> | SMMT-10B（Ours） | **40.4 / 89.5** | **27.3 / 86.9** | **28.3 / 87.1** | **28.3 / 87.4** | 33.4 / 83.0 | **32.2 / 83.4** |

---

> ### Author Response · Authors · 2025-11-26
> **(2/2) Response to Reviewer wxJV**
>
> > **[W2]** **Dependence on external components: The framework relies heavily on open-source models, such as the TTS model, which may constrain performance consistency and scalability.**
> >
>
> The core of this research lies in **exploring the performance improvement effect of speech-text fusion input on translation tasks.**
>
> We validated the effectiveness of this fusion input paradigm by comparing the use of **authentic speech** and **synthetic speech** as the audio input. Experimental results show that **synthetic speech** can similarly boost translation performance, which strongly demonstrates the inherent value of the **speech-text fusion mechanism** itself.
>
> We consider **synthetic speech primarily as a substitute in scenarios where authentic speech data is lacking.** Our main focus and contribution are on **proving that speech-text fusion significantly enhances translation task performance**, rather than on the TTS models. This finding offers a new method for applying multimodal translation in data-constrained scenarios. We believe this promblem will gradually be alleviated with the advancement of open-source TTS models.
>
> > **[W3]** **Lack of detailed ablations: It remains unclear how much each component (speech input, multimodal fusion, self-evolution) contributes to the observed performance gains.**
> >
>
> The paper presents ablation studies for the model's key components in **Tables 5 and 6**, including speech input, multimodal fusion (Table 5), and the self-evolution mechanism (Table 6).
>
> - For **speech input**：As shown in **Table 5,** synthetic speech demonstrated better performance than authentic speech on the S2TT task ($85.4$ vs. $83.4$), which is likely attributed to the advantage of having no background noise.
> - For **multimodal fusion**：As shown in **Table 5**, the difference between authentic speech and synthetic speech has a minimal impact on the overall performance of multimodal machine translation ($89.0$ vs. $89.0$). The experiments confirmed a strong semantic consistency between authentic and synthetic speech.
> - For **self-evolution mechanism**：As shown in **Table 6**, we found that after MLLM pre-training, the model's performance improved on high-resource languages. However, due to the imbalance of multilingual data, the model's performance actually decreased for some low-resource languages, such as Burmese (mya) ($88.1 \rightarrow 86.8$). To address this issue, we introduced the self-evolution mechanism, which successfully enhanced the  performance on these low-resource translation directions ($86.8 \rightarrow 88.5$).
>
> > **[Q1]** **Could the authors clarify the conceptual and methodological differences between the proposed self-evolution mechanism and reinforcement learning approaches typically used for LLM fine-tuning?**
> >
>
> Actually, this paper does not use any Reinforcement Learning methods; it belongs to Supervised Fine-Tuning.  In future research, we wil consider adding the DPO training to further improve the translation quality.

---

### Official Review · Reviewer_w9b6 · 2025-10-30

**Soundness:** 2
**Presentation:** 3
**Contribution:** 2
**Rating:** 4
**Confidence:** 4

**Summary:**

In this work, a Speech-guided Multimodal Machine Translation (SMMT) framework is proposed. The proposed system accepts textual input and synthesizes speech via the TTS model. Then, the MLLM processes both the text and synthetic speech to generate translations. The model is trained with two stages: pre-training and self-evolution training. The pre-training includes ASR to align speech and text modalities, speech translation and MMT to enable model translation capacity. The self-evolution training is a loop based training, i.e., TTS generates speech with different voices, and are assigned negative/positive tags by comparing the MT score and MMT score. Only those positive samples are used to update the MMT model. The model is evaluated with COMET score. The whole training loop is repeated if the model keeps improving COMET score.
The experimental results show the proposed system can achieve very competitive results on the MLLM dataset Multi30K, and MT dataset FLORES-200.
There are some details are missing in the paper and more detailed analysis is required. For example, what're the key factors that synthetic voice could be a positive sample or beneficial for MMT? Why speech data could help to reduce under-translation issues?

**Strengths:**

- Propose a Speech-guided Multimodal Machine Translation (SMMT) framework
- The system achieves very competitive results on multiple datasets

**Weaknesses:**

- It is understandable that information from multiple sources might help the machine learn better representation. But those features are real data instead of synthetic data. In this work, synthetic speech is used augmented MMT. Deep analysis is required to reveal the main factors that contribute the good performance. For example, we don't know the synthetic speech samples are negative or positive during inference time. What's the impact if the provided speech sample is negative one? How to select the reference voice?
- The self-evolution training could be very expensive. It is helpful to provide the training time spend on this stage.

**Questions:**

- ``We synthesize speech for the evaluation text using a fixed reference voice''. The model is updated with different voices (sec 2.4.1). How do you select the reference voice?
- what're the sizes of models in table 3?
- why speech data helps to reduce under translation?

---

> ### Author Response · Authors · 2025-11-21
> **(1/2) Response to Reviewer w9b6**
>
> ### **Thanks for the insightful comments. We mainly have updated Table 3, 4, 9, 12, 13 and Figure 5 in the revised PDF version.**
>
> > **[W1.1] It is understandable that information from multiple sources might help the machine learn better representation. But those features are real data instead of synthetic data. In this work, synthetic speech is used augmented MMT. Deep analysis is required to reveal the main factors that contribute the good performance.**
> >
>
> This paper investigates the impact of **multimodal fusion input** (text + speech) on **translation** performance. Our study shows that, compared to LLMs using only text, MLLMs processing both speech and text achieve better translation quality.
>
> The key concern is: **Why does speech improve translation performance?** This idea is inspired by the vision domain’s image-guided multimodal machine translation (**image-guided MMT**), where images help disambiguate and improve translations. Similarly, our method can be seen as **speech-guided MMT**.
>
> Existing TTS models based on diffusion models typically have accurate pronunciation and pausing. Speech is converted into **Mel-spectrograms**. Although humans cannot “read” spectrograms as easily as images, speech contains rich **prosodic features** like **stress, rhythm, and pauses**, which provide important syntactic and semantic cues. These help the model reduce ambiguity, highlight key information.
>
> To validate this, as shown in the Table, we compared two speech inputs: TTS speech with **full prosody**, and synthesized speech concatenating every four words (**partial prosody**). Results show that even weakened prosody beats text-only, and **full prosody TTS speech performs best.**
>
> | spBLEU / COMET | **SMMT （full prosody TTS speech + Text）** | **SMMT (partial  prosody TTS speech + Text)** | **Baseline (text-only)** |
> | --- | --- | --- | --- |
> | eng_jpn | **35.2 / 92.7** | 34.9 / 91.9 | 33.3 / 91.3 |
> | eng_kor | 30.1 **/ 90.5** | 30.3 / 90.1 | 30.4 / 90.1 |
> | eng_cmn | **42.6  / 91.2** | 42.5 / 89.4 | 41.6 / 89.2 |
>
> > **[W1.2]** **For example, we don't know the synthetic speech samples are negative or positive during inference time. What's the impact if the provided speech sample is negative one?**
> >
>
> Generally, the **speech quality of existing** **TTS models for synthesizing short texts is controllable**. However, failure cases (or negative samples) do exist, such as when the input text is excessively long. When the input text is relatively long, existing TTS models are prone to introducing noise (such as word omissions or generating audio exceeding 30 seconds). As shown in Table 4, we also conducted experiments on the WMT24++ dataset. Although the model's performance on the overall dataset is comparable, it exhibits best performance for input text within the **$<200$** range.
>
> More importantly, as shown in Table 4, the model's performance does not significantly degrade compared to the baseline (33.4 / 83.0 vs. 33.9 / 82.7), even when receiving noisy speech input. This fully demonstrates the model's robustness.
>
> Averaging Text Length: Flores-200 (130.4), WMT24++ (191.3).
>
> | **spBLEU / COMET** | FLORES-200 (eng → 27) | FLORES-200 (jpn → 27) | FLORES-200 (kor → 27) | FLORES-200 (cmn → 27) | **WMT24++ (eng → 22)** | **WMT24++ (eng → 22 (<200))** |
> | --- | --- | --- | --- | --- | --- | --- |
> | DeepSeek-V3.1-671B | 39.3 / 88.9 | 26.1 / 85.7 | 27.7 / 85.9 | 27.5 / 86.2 | 34.1 / **83.6** | 31.8 / 83.4 |
> | Gemma3-27B-it | 37.4 / 88.0 | 23.8 / 81.0 | 25.0 / 81.2 | 24.5 / 81.5 | **34.3** / 82.9 | 31.8 / 82.6 |
> | NLLB-moe-54B | 35.7 / 86.3 | 21.8 / 81.7 | 23.6 / 83.7 | 22.8 / 82.1 | 25.4 / 76.9 | 24.4 / 77.7 |
> | Qwen3-Next-80B-A3B | 34.5 / 86.6 | 22.9 / 83.8 | 23.9 / 83.9 | 24.2 / 84.3 | 30.5 / 81.5 | 29.6 / 81.6 |
> | Baseline (text-only) | 39.7 / 88.3 | 26.6 / 85.4 | 27.4 / 85.6 | 27.5 / 85.7 | 33.9 / 82.7 | 32.1 / 82.9 |
> | SMMT-10B（ours） | **40.4 / 89.5** | **27.3 / 86.9** | **28.3 / 87.1** | **28.3 / 87.4** | 33.4 / 83.0 | **32.2 / 83.4** |
>
> > **[W1.3]** **How to select the reference voice?**
> >
>
> In the experiments, we provide the user with a high-quality reference speech sample by default based on `CosyVoice2`, and perform cloning based on this to ensure stable performance.

---

> > ### Comment · Reviewer_w9b6 · 2025-11-27
> > **Comments based on rebuttal**
> >
> > It is appreciated that authors provided detailed replies for questions I mentioned. Most questions are answered and I'd like to raise the score.
> > One more question, in [W1.2]  I understand TTS will generate bad utterances. My main question how do we identify positive and negative samples during inference time. However, I think the reply doesn't address this directly. Instead,
> > "More importantly, as shown in Table 4, the model's performance does not significantly degrade compared to the baseline (33.4 / 83.0 vs. 33.9 / 82.7), even when receiving noisy speech input. This fully demonstrates the model's robustness."
> > Does that mean it is ok to use negative samples during inference time?

---

> ### Author Response · Authors · 2025-11-26
> **(2/2) Response to Reviewer w9b6**
>
> > **[W2]** **The self-evolution training could be very expensive. It is helpful to provide the training time spend on this stage.**
> >
>
> The self-evolution mechanism was introduced in the experiments to balance the issue of insufficient training data for real speech in **low-resource language directions**. For example, the **FLEURS dataset has only less 3,000 train samples.** Therefore, during the self-evolution process, the actual number of synthesized speech samples is very small, and the GPU requirements are very low.
>
> For instance, with an 8-card setup with 8GB GPUs, it only takes less than **30 minutes** to synthesize **3,000 samples**, so the actual computational load is not high. Since the number of samples is very small, the training time is not long.
>
> > **[Q1]** **We synthesize speech for the evaluation text using a fixed reference voice. The model is updated with different voices (sec 2.4.1). How do you select the reference voice?**
> >
>
> During training, we aim to use as many speech samples as possible to enhance robustness, while during inference, we provide the user with a high-quality reference speech sample by default, and perform cloning based on this to ensure stable performance. All evaluations can be conducted using the reference audio provided for this experimental verification.
>
> > **[Q2]** **What're the sizes of models in table 3?**
> >
>
> | **Model** |  |
> | --- | --- |
> | DeepSeek-V3.1 | 671B |
> | Gemma3-27B-it | 27B |
> | NLLB-moe-54B | 54B |
> | Qwen3-Next-80B-A3B | 80B |
> | WRA-guided、RG-MMT-EDC、Soul-Mix、CONSqa-MMT、VALHALLA、Bridge | <1B |
> | DreamLLM、IMAGE | 7B |
> | Baseline  | 9B |
> | SMMT (Ours) | 10B |
>
> > **[Q3]** **Why speech data helps to reduce under translation?**
> >
>
> We found that the improvement observed in some human recheck cases stems from the resolution of **under-translation** issues. As shown in **Figure 5**, the MLLM, having undergone pre-training on speech tasks, learned the alignment between speech and text. Subsequently, receiving a **fused input of speech and text**, it became less likely to overlook the text in the input, thereby reducing omission errors.

---

> ### Author Response · Authors · 2025-11-28
>
> Thank you for your insights. We appreciate you raising the score.
>
> During the inference phase, it is difficult to determine whether a sample is positive or negative. Therefore, we included **experiments that accept negative samples**.
>
> Specifically, we further conducted comparative experiments on both **short** and **long** **texts (negative)**. Based on the WMT24++ dataset, our experimental design yielded the following findings:
>
> **Experimental Findings:**
>
> 1. **Short Texts (<200)**: For existing TTS models, when the input text length is less than 200, the quality of the synthesized speech is generally acceptable, resulting in a **significant performance improvement** (32.1 / 82.9 vs. **32.2** / **83.4**).
> 2. **Long Texts (>500) - Extreme Case**: We investigated extreme cases where the input length is entirely **greater than 500**. In this scenario, the synthesized speech exhibits **obvious negative sample noise**, such as excessive speed, missed words, or being overly long（>30s）.
>     - In this extremely adverse situation, even with input lengths greater than 500, the model's performance shows **no significant drop** compared to the baseline (e.g., 33.6 / 83.0 vs. **33.7** / 82.9).
>     - This is attributed to the **fusion of speech and text inputs**; even if the synthesized speech contains some noise, the final performance does not deviate substantially from the pure text performance.
>
> Therefore, we conclude that the translation quality remains **acceptable** even when **negative samples are utilized**.
>
> | **spBLEU / COMET** |  | **WMT24++ (eng → 22 )** |  |
> | --- | --- | --- | --- |
> | Input Text Length | ALL  | <200  | **>500 (negative)** |
> | Baseline (text-only) | 33.9 / 82.7 | 32.1 / 82.9 | 33.6 / 83.0 |
> | SMMT-10B（TTS speech + Text） | 33.4 / **83.0** | **32.2 / 83.4** | **33.7** / 82.9 |

---

### Official Review · Reviewer_s3NW · 2025-10-31

**Soundness:** 3
**Presentation:** 3
**Contribution:** 3
**Rating:** 8
**Confidence:** 4

**Summary:**

This paper addresses Speech-guided Multimodal Machine Translation (SMMT), i.e., improving text translation by providing semantically equivalent speech at the input. During evaluation, this "missing" speech is generated by TTS.
The model uses a Whisper speech encoder which is frozen, a speech adapter based on Q-Former and an MLP, and a strong multilingual LLM based MT system (GemmaX2-28-9B) which is LoRA adapted.

The approach uses a multistage curriculum pretraining with progressively complex objectives: ASR, S2TT and SMMT. The authors also propose a "self-evolution algorithm", i.e. iteratively complementing the training data with synthesized speech while making sure that the additional speech improves performance.

The model is compared with other multimodal MT system which use the image modality (authentic and synthetic images) on the Multi30k benchmark. It performs best for 5 out of 6 tasks. The model also outperforms SOTA on Flores200, but the model is only test on 28 languages. Also, the base MT system, i.e., to which speech input is added, already outperforms the other systems for 5 out of 6 languages.

**Strengths:**

It's interesting to see that adding synthetic speech to the input of a text translation system improves performances. This seems to be more effective than adding images to the input (cf. Table 3).

The incremental training strategy is interesting and I wonder whether similar techniques could be used in other LLM settings than MT.

**Weaknesses:**

The experimental results show that the method works, but the authors fail to justify, or to try to explain, why it works! The only argument is that the prosody in speech helps translation. I could agree with this if human speech were used. However, this work addresses the issue how to improve text translation by providing in addition *synthetic speech*. I am not well aware of the current SOTA in TTS, but I would be surprised that the prosody is very rich.

It is not clear which data was used to train the speech and LLM adapter. The data mentioned in Table 9 gives no data sizes. I wonder whether the observed improvements are mainly the result of the additional training data. An important ablation would be to LoRa adapt the MT systems on the data, but without speech input.

In Table 4 and 12, good improvements are reported for several languages when translating from 4 languages into 27 foreign languages. Given that the algorithmic improvements are on the input only, I am puzzled how we can achieve a large variance on the improvements when translating from the same language into many different ones. The paper would benefit from a deeper analysis, instead of numerical metrics only.

I would appreciate some examples, i.e., some sentences from Flores that are better translated with speech augmented input.

**Questions:**

- it's unclear on what data the model was trained on. Table 9 mentions Fleurs train, Common Voice train and Multi30k, but no sizes (per language) are given.

- why you support 28 languages only? Is this limited by the TTS system? What type of training data do you need to scale to more languages?

- the gains form the baseline text-only MT system and the proposed SMMT-10B are much higher for FLICK and MSCOCO than Flores. How can you explain this? What are the characteristics of the sentences in FLICK/MSCOC, e.g. very short, caption like?

- Table 9 caption: "we removed overlapping portions from the FLORES devtest set". You report results on Flores devtest (cf. Table 10). You should never modify a test set! This makes it impossible to compare results. Which version of Flores was used in Tab 4, your modified one? Did you recalculate the scores for the other systems?

---

> ### Author Response · Authors · 2025-11-21
> **Response to Reviewer s3NW**
>
> ### **Thanks for the insightful comments. We mainly have updated Table 3, 4, 9, 12, 13 and Figure 5 in the revised PDF version.**
>
> > **[W1]** **The experimental results show that the method works, but the authors fail to justify, or to try to explain, why it works! The only argument is that the prosody in speech helps translation. I could agree with this if human speech were used.  I am not well aware of the current SOTA in TTS, but I would be surprised that the prosody is very rich.**
> >
>
> In our experiment, we used **`CosyVoice2`** as the **TTS model**, which, as shown in the paper[1], 'CosyVoice demonstrated high **prosody** naturalness, content consistency, and speaker similarity in speech in-context learning.'
>
> We found that the improvement observed in some human recheck cases stems from the resolution of **under-translation** issues. As shown in **Figure 5**, the MLLM, having undergone pre-training on speech tasks, learned the alignment between speech and text. Subsequently, receiving a **fused input of speech and text**, it became less likely to overlook the text in the input, thereby reducing omission errors.
>
> > **[W2]** **It is not clear which data was used to train the speech and LLM adapter. The data mentioned in Table 9 gives no data sizes. I wonder whether the observed improvements are mainly the result of the additional training data. An important ablation would be to LoRa adapt the MT systems on the data, but without speech input.**
> >
>
> We have **added** the results of the LoRA training of the base model to **Table 3**. Because the datasets for **speech-to-text translation** are relatively **limited**, the improvement from using the data for machine translation **LoRA training** is limited.
>
> Furthermore, we have included the size of the speech data in **Table 9**.
>
> > **[W3]** **In Table 4 and 12, good improvements are reported for several languages when translating from 4 languages into 27 foreign languages. Given that the algorithmic improvements are on the input only, I am puzzled how we can achieve a large variance on the improvements.**
> >
>
> These inconsistent improvements result from language **resource disparities** due to the introduction of speech data and the self-evolution mechanism. As shown in **Figure 3**, specifically:
>
> - For **Low-Resource Languages**, high-quality speech information or synthetic data acts as a form of data augmentation, potentially leading to more significant performance gains.
> - For **High-Resource Languages**, the baseline model already performs well, meaning the Marginal Gain provided by the supplementary speech information is smaller, and the improvement in magnitude is correspondingly lower.
>
> > **[W4] I would appreciate some examples.**
> >
>
> We **added** **Figure 5** to the revised paper to show examples.
>
> > **[Q1] It's unclear on what data the model was trained on in Table 9.**
> >
>
> We have **updated Table 9** with the approximate total duration of speech training data for each dataset.
>
> > **[Q2] Why you support 28 languages only? Is this limited by the TTS system? What type of training data do you need to scale to more languages?**
> >
>
> Support for 28 languages is determined by our chosen LLM foundation model, **`GemmaX2-9B`**, which is reported to support 28 languages in its paper [2].
>
> In our latest study, we found that adding **50-100 hours of ASR data** and **10 hours of S2TT data** for per augmented language can achieve comparatively good results. This data is typically available from the Common Voice and Fleurs datasets.
>
> > **[Q3] The gains form the baseline text-only MT system and the proposed SMMT-10B are much higher for FLICK and MSCOCO than Flores. How can you explain this?**
> >
>
> The difference primarily stems from two reasons:
>
> First, as shown in **Table 3** and **Table 4**, **the input length** in the Multi30k dataset are significantly shorter (averaging **59.3**) compared to those in Flores-200 (averaging **130.4**). Our model performs better with relatively shorter texts.
>
> Second, our model was trained on the Multi30k dataset and was **validated on the validation set**. The model could more easily learn the data distribution specific to this dataset.
>
> > **[Q4] Table 9 caption: "we removed overlapping portions from the FLORES devtest set". You report results on Flores devtest (cf. Table 10). You should never modify a test set!**
> >
>
> We have **updated Table 9** in the revised paper. There is a **misunderstanding** here. We did not modify the Flores test set. Instead, we **removed the overlapping portion** of the **Fleurs training dataset** to prevent data leakage. We apologize for the error in the description of the previous version.
>
> ### **References**
>
> [1]: Du Z, Wang Y, Chen Q, et al. Cosyvoice 2: Scalable streaming speech synthesis with large language models[J]
>
> [2]: Cui M, Gao P, Liu W, et al. Multilingual Machine Translation with Open Large Language Models at Practical Scale: An Empirical Study[C]

---

### Author Response · Authors · 2025-11-26
**A Friendly Reminder: A Polite Follow-Up Regarding Rebuttal**

Dear reviewers,

We understand you may be very busy, and we truly appreciate your time and consideration. We want to ensure that we have addressed all your concerns satisfactorily. If there are any points we may have overlooked or any inconsistencies you would like us to clarify, we would greatly appreciate your guidance.

With sincere appreciation,

All authors

---

### Author Response · Authors · 2025-11-30
**Rebuttal Summary**

We sincerely thank all reviewers for their thoughtful feedback and constructive suggestions. In our detailed responses, we have endeavored to address all major concerns through additional clarifications, theoretical discussions, and supplementary experiments. We also made targeted revisions to the manuscript.

- **Reviewer s3NW — Rating: 8, Confidence: 4** (No Response)
- **Reviewer w9b6 — Rating: 4 → ≥6, Confidence: 4** *“It is appreciated that authors provided detailed replies for questions I mentioned. Most questions are answered and I'd like to raise the score.”*
- **Reviewer wxJV — Rating: 4, Confidence: 2** (No Response)

Based on the responses received, **only Reviewer w9b6 has responded and indicated that they raised their score**. Notably, Reviewer **wxJV** gave a score with a **confidence of 2**. Below, we summarize the key strengths highlighted by the reviewers and our responses to the main concerns.

> **Strengths**
>

We thank the reviewers for highlighting the key strengths of our work:

- This paper proposes a **Speech-guided Multimodal Machine Translation (SMMT)** framework. The approach effectively unifies speech and text modalities and extends to multiple languages, showing potential for broader generalization.
- **Innovative Self-Evolution Mechanism:** The framework introduces a novel self-improving incremental training strategy that leverages translation performance metrics as optimization objectives, facilitating continuous enhancement through iterative evolution cycles.
- **It's interesting to see that adding synthetic speech to the input of a text translation system improves performance.** The method achieves very competitive results on multiple datasets.

> **Response for Main Concerns**
>

We have carefully addressed the main concerns raised by the reviewers:

**(1) A Deep Analysis Is Required for the Good Performance.（Reviewer s3NW, w9b6）**

- **Mechanism:** Our method functions as **Speech-guided Multimodal Machine Translation**, similar to how image-guided MMT uses visual information for disambiguation; it significantly improves translation performance by providing **prosodic features** as auxiliary cues. Receiving a **fused speech-and-text input** makes the model less prone to overlook the text, thereby reducing under-translation errors.
- **Validation:** We compared two speech inputs: TTS speech with **full prosody** and synthesized speech concatenating every four words (**partial prosody**). Results confirm that even weakened prosody outperforms text-only input, and **full prosody TTS speech performs the best**, validating the importance of prosodic features.

**(2) Evaluation Scope** **& Ablation Study (Reviewer s3NW, wxJV)**

- **Evaluation Expansion:** Experiments were conducted on Multi30K, FLORES, and the newly **added WMT24++ datasets** (Table 4), covering common multimodal and widely-used multilingual machine translation benchmarks.
- **Ablation Study:** We performed key ablation studies focusing on four modules: **LoRA Contribution** (Table 3), **Speech Input** (Table 5)**, Multimodal Fusion** (Table 5)**, and Self-Evolution** (Table 6).

**(3) Training Data Details and the Training time for** **Self-Evolution (Reviewer  s3NW, w9b6)**

- **Training Data Details:** **Table 9 has been updated** to include the approximate total duration of the speech training data for each dataset.
- **Self-Evolution Time:** The **Self-Evolution** mechanism was introduced to address insufficient speech data in **low-resource language directions** (e.g., FLEURS dataset with $<3,000$ training samples). Synthesizing 3,000 samples on an 8-card setup (8GB GPUs) takes **less than 30 minutes**.

**(4) Multilingual Performance & Case Study (Reviewer s3NW)**

- **Multilingual Performance:** Inconsistent improvements stem from **language resource disparities**. For **Low-Resource Languages**, speech acts as potent **data augmentation**, yielding **more significant performance gains**.
- **Case Study:** **Figure 5** has been added to the revised paper, providing concrete examples illustrating these effects.

**(5) External Dependence** **(Reviewer wxJV)**

- The core of this research is to explore the performance improvement effect of **speech-text fusion input** on translation tasks. The study primarily focuses on validating the effectiveness of the fusion input, treating **synthetic speech** as a **substitute** in inference scenarios where **authentic speech data is lacking**. Furthermore, we think that the issues of performance consistency and scalability will be alleviated with the advancement of open-source TTS models.

---

> ### Author Response · Authors · 2025-12-03
> **Clarifications on Initially Negative Scores**
>
> To clarify the two initially negative scores, which stem largely from misunderstandings rather than substantive flaws, we provide the following focused explanations:
>
> - **Reviewer w9b6:** Although the initial rating was 4, the reviewer **acknowledged the effectiveness of speech-text fusion** for improving translation tasks. Their concerns focused on the **analysis of synthetic speech** for augmented MMT and the **efficiency** of the self-evolution mechanism. We addressed each point in detail to clarify the misunderstandings, after which the reviewer **agreed to raise the rating**.
> - **Reviewer wxJV**: Although the initial rating was 4, **the reviewer gave good scores for both Soundness and Contribution**. Their concerns stemmed primarily from the **evaluation scope, ablation studies,** and **the dependence on TTS models**. We **added results on new datasets** and provided **comprehensive ablation analyses**. Our mechanism is **effective with both** **authentic and synthetic** speech. We noted that the reviewer assigned **a low confidence score of 2** and **did not participate in the rebuttal discussion**. We hope that our response and additional experiments have effectively clarified the misunderstandings.

---

### Meta-Review · Area_Chair_jGtF · 2026-01-20

**Summary:**

Before rebuttal phase, the reviewers’ main pre-rebuttal concerns were:
(1) insufficient explanation/analysis for why speech-text fusion helps (and whether gains come simply from more synthetic data), (2) missing key ablations and details (e.g., speech vs no-speech on matched data, module-wise ablations, model-size info, and a potentially confusing statement about FLORES), and (3) questions about practicality/generalization, including dependence on external TTS at inference, robustness to noisy/“negative” synthetic speech, and the compute cost/distinction of the self-evolution procedure.

**Reviewer Concerns:**

The author have addressed most of the reviewers' questions, including

(1) “why it works” (s3NW, w9b6): Authors provided a clearer hypothesis and added supporting evidence (including a full-prosody vs degraded/partial-prosody comparison, and qualitative examples).

(2) Key ablations (s3NW, wxJV): Added/expanded ablations separating (i) LoRA contribution, (ii) speech input, (iii) fusion, and (iv) self-evolution.

(3) Evaluation scope (wxJV): Expanded to additional datasets beyond the original set (including a WMT-style benchmark) and added more translation direction

(4) Inference-time “negative speech” concern (w9b6): Added robustness results suggesting performance does not materially degrade even when TTS is noisy (including an “extreme long input” setting).

(5) Dependence on external TTS at inference (wxJV): Even if robustness is demonstrated, the approach still relies on a TTS component; the rebuttal frames this as acceptable and improving with better open-source TTS, but it remains a structural dependency.

**Reviewer Scores:**

Reviewer w9b6: 4 → predicted 6. This reviewer explicitly signaled they would raise the score after seeing detailed answers. With full participation, they likely land at a borderline accept (around 6).

Reviewer wxJV: 4 → predicted 6. Given the reviewer’s low confidence and that many of their stated issues (evaluation scope + ablations) were directly addressed, full participation would likely move them upward.

Therefore, I think this paper can be confidently accepted.

---

### Decision · Program_Chairs · 2026-01-26

Accept (Poster)